# Impact of Foehn Wind and Related Environmental Variables on the Incidence of Cardiac Events

**DOI:** 10.3390/ijerph17082638

**Published:** 2020-04-12

**Authors:** Andrzej Maciejczak, Agnieszka Guzik, Andżelina Wolan-Nieroda, Marzena Wójcik, Teresa Pop

**Affiliations:** 1Department of Neurosurgery, Saint-Luke Hospital, 33-100 Tarnów, Poland; amac@mp.pl; 2Institute of Medical Sciences, Medical College, University of Rzeszów, 35-959 Rzeszów, Poland; 3Institute of Health Sciences, Medical College, University of Rzeszów, 35-959 Rzeszów, Poland; wolan.a@gmail.com (A.W.-N.); marzena.88@poczta.fm (M.W.); popter@interia.pl (T.P.)

**Keywords:** myocardial infarction, foehn wind, halny, environmental variables

## Abstract

In Poland there is no data related to the impact of halny wind and the related environmental variables on the incidence of cardiac events. We decided to investigate the relationship between this weather phenomenon, as well as the related environmental variables, and the incidence of cardiac events in the population of southern Poland, a region affected by this type of wind. We also decided to determine whether the environmental changes coincide with or predate the event examined. We analysed data related to 465 patients admitted to the cardiology ward in a large regional hospital during twelve months of 2011 due to acute myocardial infarction. All the patients in the study group lived in areas affected by halny wind and at the time of the event were staying in those areas. The frequency of admissions on halny days did not differ significantly from the admissions on the remaining days of the year (*p* = 0.496). No statistically significant differences were found between the number of admissions on halny days and on the remaining days during halny months (*p* = 0.084). We have identified a difference in the number of admissions between days with no halny and days immediately preceding onset of halny (*p* = 0.001). However, no effects of the related environmental variables have been observed in the incidence of cardiac events (*p* = 0.866, F = 0.37). On the days with halny wind, incidence of cardiac events is similar to that on the remaining days of the year.

## 1. Introduction

Cardiovascular disorders are the most common cause of mortality worldwide, including Poland and most European countries [1]. Seasonal peaks in acute myocardial infarction incidence have been widely reported. Weather has been postulated to be one of the elements greatly contributing to this association [2]. However, it is difficult to clearly establish relationships between meteorological or synoptic factors and decline in people’s mood or well-being. The reasons for that may lie, e.g., in varied responses to weather related stimuli, or in synergistic effects produced by various meteorological factors. Due to this, results obtained by various researchers frequently cannot be compared because of the different methodologies applied in studies. Nevertheless, in order to get better understanding of health-related effects produced by various environmental factors, there is a need for further research focusing on possible links between specific medical conditions and selected weather phenomena.

The term “föhn” is used with reference to a type of strong and turbulent wind blowing down from mountain crests towards the lee side and bringing dry and warm air to the surface. Foehn events rather frequently occur in the Alps, or more specifically on the northern side of the mountains, and constitute a significant challenge in weather forecasting for the Alpine valleys, inhabited by many people. The northern Alpine foehn is frequently observed in the prefrontal environment. The foehn clearance reflects the leeward range of the foehn in the mountains, i.e., a region with no clouds where dry and warm air occurs at the surface, just as in the foehn region but with no strong winds. This meteorological phenomenon is not unique to the Alps, and it can be observed over other high mountains worldwide [3].

A local variety of foehn, “halny” wind, is produced by violent flow of air masses from the southern side of the Carpathians or the Sudetes. This happens when Poland is approached from the west by a deep cyclone (low-pressure area), while the anticyclone centre (high-pressure) is in Ukraine [3,4]. In addition to an abrupt change in temperature and its large speed, there are also other consequences of halny wind. The first involves rapid pressure drop, and a second is linked with increased condensation of positive ions in the air as a result of which microscopic particles of gases and aerosols, including harmful elements and chemical compounds, can easily penetrate organisms [3,4]. Halny wind frequently leads to considerable increase in the temperature and decrease in humidity on the leeward side of the mountains. The strong abrupt gusts of wind at times reach a speed typical of a hurricane and are accompanied with characteristic calm intervals. The foehn winds are observed in the Tatras rather frequently, particularly during the cold seasons. The criteria commonly applied by Polish researchers investigating foehn winds occurring in the mountainous areas include: direction (usually 90–270°) and wind speed (>4 m/s, commonly ≥10 m/s, at times increasing to even 60 m/s), occurrence of strong gusts and calm intervals, sudden rise in the air temperature (by several degrees in a short time) and decrease in the humidity (commonly ≤70%), and changes in atmospheric pressure and/or certain types of cloud. In foothill and lowland areas, the wind is dry and warm [4].

It is believed that foehn wind may lead to increased aggressiveness, more severe depressive states, emotional crises ending with suicide attempts, agitation, anxiety, fatigue, weariness, muscle and joint pain, problems with concentration, sleep disorders, migraine, and headaches [4,5,6,7].

People with high sensitivity to weather changes are referred to as meteoropaths. They present increased tension towards the vegetative system and poor adaptative responses. Effects of weather fronts, in particular cold fronts, include decrease in the temperature and increase in the atmospheric pressure, responses to which include vasoconstriction, potentially posing a burden for the circulatory system [8,9,10]. Nowadays, it is estimated that the number of meteoropaths is consistently growing. According to the data from the Institute of Meteorology and Water Management, as many as 50–70% of Poles respond to weather changes [11]. Similar statistics have been published in other countries, e.g., in Germany 54.5% of the population is weather sensitive and in Canada the relevant rate amounts to 61% [12]. It is also believed that meteoropathy is a “disease” of affluence, caused by stress, work overload, and unhealthy lifestyles, since meteoropaths are mainly residents of urban areas, whose immunity is affected by the long time spent in the artificial microclimates of their homes and offices, by a lack of exercise and by polluted air. It has also been pointed out that meteoropathy is most commonly observed in individuals affected by coronary disease, hypertension and ulcer disease [8,9,10,13].

A large body of evidence reported in the literature shows there is an association between wind- related environmental variables and cardiovascular or cerebrovascular events, with impact observed also on the preceding days [14,15,16,17,18,19,20,21,22,23]. Boussoussou et al. have reported positive significant association between acute cardiovascular diseases and a cold front effect lagged by one day [14]. Mohammad et al. demonstrated that incidence of myocardial infarction increased with higher wind velocity, lower air temperature, lower atmospheric air pressure, and shorter sunshine duration [15]. Similarly, Honda et al. found that increased incidence of acute myocardial infarction corresponds to lower minimum temperature occurring on the second day preceding the onset of wind [17]. Radišauskas et al. established that myocardial infarction morbidity is directly correlated to wind speed and atmospheric pressure and inversely correlated to atmospheric temperatures and rainfall level [16]. Goerre et al. reported that the incidence of myocardial infarctions is significantly affected by wind activity, ambient pressure, and pressure gradient [15]. Vencloviene et al. showed that prognostic factors for the risk of acute coronary syndromes include such environmental variables as wind speed, daily air temperature, and barometric pressure on the day of the onset and on the two days preceding the ambulance call [18]. Likewise, Ezekowitz et al. showed relationships between the likelihood of an acute coronary event and rapid changes in weather conditions (including excessively frequent increases in wind speed) [19]. Hensel et al. showed that increasing wind speed corresponds to greater likelihood of out-of-hospital cardiac arrest of presumed cardiac aetiology [20]. Vencloviene et al. demonstrated that increased risk of paroxysmal atrial fibrillation occurring at night was linked with wind speed above the median (lag 2–4 days) [21]. Tamasauskiene et al. pointed out that a greater risk of ischemic stroke in older patients was associated with high wind speed on the day of the incident [22]. Similarly, Kim et al. showed that the odds ratios for wind speed, daily wind speed range, and wind chill for patients with ischemic stroke were slightly greater than in the total sample [23].

Effects of seasonal changes in environmental variables have also been pointed out. According to Gallerani et al., it is possible to observe six-month-period regularities in the incidence of subarachnoid haemorrhage, spring and autumn being the peak seasons. The authors propose an explanation that during these transitional seasons the arterial wall is exposed to frequent stress episodes due to typical changes in temperature and that may result in aneurysmal rupture occurring more frequently than, e.g., during the more stable periods of winter cold [24]. Investigators from the International Worldwide Registry reported that the high incidence of acute aortic dissection in winter is observed in various climate settings (both cold and temperate), which seems to suggest that the major mechanistic factors contributing to the phenomenon include the relative change in temperature, rather than absolute temperature, and/or circannual rhythms [25].

Clinical studies provide evidence that some meteorological factors, and most importantly weather changes occurring rapidly, e.g., in one day, may play an important role in inducing subjectively perceived indisposition in healthy individuals and in promoting disease symptoms in patients. Notably, the same weather changes may induce completely different bodily responses. As an example, a large drop in barometric pressure will induce a decrease in blood pressure in some individuals and an increase in other people [2,26,27,28,29]. Due to this no uniform recommendations for dealing with sudden changes in weather can be addressed, e.g., to all patients with cardiovascular problems.

It has been shown that meteorotropic responses in both healthy people and individuals affected by medical conditions may be observed more in advance of specific changes in weather; in extreme cases these may be “sensed” up to 48 h before they actually happen. Such sensitivity may result from the changes in the electric field of the atmosphere, preceding arrival of a weather front [30,31,32]. It has also been reported that changes in atmospheric pressure produce adverse effects in the human body, possibly also leading to cardiac incidents [33,34,35,36].

As early as 1953, it was suggested by Petersen that the human organism may be stimulated by specific relationships between weather fronts [6]. Cardiovascular problems are most frequently listed among meteorotropic diseases. In Poland the earliest studies focusing on impact of weather on functioning of the cardio-vascular system were carried out in the mid-1800s [6]. Research shows impact of short-term weather changes on intensity of ischemic heart disease symptoms, increased oscillations in arterial blood pressure, and higher incidence of acute myocardial infarction. Over 34% of fatal cases of myocardial infarction co-occur with a passage of weather fronts, compared to only 6% observed during periods of calm weather linked with high pressure [37]. A survey conducted by Baranowska showed that in the respondents’ opinion, meteoropaths are adversely affected by weather transitions from highs to lows [6]. In a study conducted by Höppe, individuals sensitive to weather, living in northern Germany, negatively assessed stormy weather (30%) and influx of cold air (29%). On the other hand, inhabitants of southern Germany experienced mood decline accompanying influx of warm (30%) rather than cold air (18%). This difference may be associated with the quite frequent foehn winds in southern Germany [38]. Mackensen et al. investigated effects of weather in the sense of well-being observed in people living in Germany and in Canada. The authors reported weather sensitivity in 54.5% of the subjects representing the German population while in Canada the rate amounted to 61% [12]. Likewise, Shiue et al. found a relationship between weather conditions, including wind speed, and hospital admissions due to heart disease in Germany [39]. The same relationship was shown by studies conducted in Switzerland [40] and Lithuania [16].

The foehn wind known as “halny”, occurring in southern Poland, has as equally bad a reputation as the foehn winds blowing in other parts of Europe and the world. The aim of this study was to examine the impact of the foehn wind and the related environmental variables on the incidence of cardiac events in the population inhabiting a region affected by halny wind. We also decided to determine whether the environmental changes coincide with or predate the event examined.

## 2. Materials and Methods

### 2.1. Participants and Procedures

We examined a consecutive series of patients with acute myocardial infarction classified according to ICD-10 in the medical records as I21 (the whole subgroup of I21.0—I21.4 and I21.9), admitted to the cardiology ward in a large regional hospital during the twelve months of 2011 [41]. Patients taken into account in the study presented evident symptoms of heart disease which started within 12 h before admission. Patients with similar symptoms which began earlier than 12 h prior to admission were not included in the study group. The criteria used in the diagnosis of acute myocardial infarction included: (1) STEMI (ST-elevation myocardial infarction); (2) NSTEMI (non ST-elevation myocardial infarction) with elevated troponin levels, exceeding the norm adopted by our hospital; the elevated concentrations were confirmed in two consecutive tests carried out 3–4 and 9–12 h after the first measurement. The diagnosis of acute myocardial infarction in the majority of the patients (65%) was based on the first criterion. Troponin levels were also measured in the patients with STEMI, as a way to monitor progress of the infarction. All the patients in the study group lived in areas affected by halny wind and at the time of the event were staying there. The hospital is also located in an area affected by halny.

Data related to the weather conditions during the 365 days of year 2011 were obtained from the Institute of Meteorology and Water Management—National Research Institute in Warsaw and from meteorological stations no. 11934 in Poprad, 12650 on Mount Kasprowy Wierch and 12566 in Kraków, all situated in or near the Tatra mountains [42].

Descriptions of weather conditions (wind direction and speed) as well as synoptic charts were taken into account to identify days with halny wind blowing.

### 2.2. Ethics

The study was reviewed and approved by the Bioethics Commission of the Medical Faculty at University of Rzeszow. All procedures were carried out in full compliance with the Declaration of Helsinki.

### 2.3. Data Analyses

We examined the number of cardiac incidents occurring: (i) on the days with halny wind blowing; and (ii) on days immediately preceding onset of halny. We compared these numbers to those occurring on the remaining days of the year. The statistical analyses were performed using Calc programme. Chi-squared test was applied to compare the frequency of the specific phenomenon occurring relative to the patients’ sex, in order to confirm or refute the effect of this variable. The test was used because of the qualitative nature of the data. Frequency of one phenomenon coinciding with another phenomenon was examined with two-tailed test for significance of structure indicators. Temporal correlations between admissions and weather events caused by halny were analysed using Pearson’s correlation coefficient. Additional relationship between environmental variables and number of cardiac events was examined using multiple regression analysis. Changes in mean daily air temperature, atmospheric humidity, speed of wind, pressure and total daily precipitation were shown in linear graphs. Statistical significance was assumed at *p* < 0.05.

### 2.4. Sample Size

The minimum size of the sample was determined using a sample size calculator, taking into account the number of individuals admitted to the cardiology ward due to acute myocardial infarction, i.e., approximately 465 patients. A fraction size of 0.8 was used, with a maximum error of 5%, and a sample size of 161 patients was obtained. In our study we analysed data related to 227 individuals admitted on halny days, one day before onset of halny and during the remaining days of the relevant months.

## 3. Results

### 3.1. Epidemiology

In 2011 a total of 465 patients were admitted to the cardiology ward due to acute myocardial infarction. The group included 168 females (36.1%) and 297 males (63.9%). The number of male patients admitted was significantly higher compared to the female patients. This fact was confirmed in the test for significance (*p* < 0.001) (Table 1).

### 3.2. Halny Wind Occurrence

In 2011 Halny wind was observed in the months of January, March, October, November, and December. On the whole, during those months halny occurred nine times and blew for a total of 22 days (Table 2).

### 3.3. Number of Admissions in Months with Halny Wind and Patients’ Sex

No statistically significant relations were found between the patients’ sex and number of admissions on the halny days (*p* = 0.46) (Table 3). Females hospitalised due to myocardial infarction on halny days accounted for 13.2% and on days without halny for 86.8% of all admissions. Males admitted on windy days accounted for 9.9%, and on days with no wind for 90.1% of all those admitted. Admission rate for females did not differ significantly from admission rate for males on halny days and on the remaining days.

### 3.4. Number of Admissions on Halny Days and on the Remaining Days of the Year

Frequency of admissions on halny days did not differ significantly from the admissions on the remaining days of the year. The *p* value for the difference between the two structure indicators amounted to *p* = 0.496—Table 4.

Table 5 shows results of statistical analyses taking into account the months with halny wind. No statistically significant differences were found between the number of admissions on halny days and on the remaining days during halny months—*p* = 0.084.

On the other hand, we have identified a difference in the number of admissions between days with no halny and days immediately preceding onset of halny. During the 24 h immediately preceding halny wind, the number of admissions was significantly higher than on the remaining days of months with Halny (Table 5).

### 3.5. Temporal Correlation between Admissions and Weather Events Caused by Halny

No statistically significant correlations were found between the environmental variables and number of cardiac events in any of the periods investigated. The analyses took into account a period of five months during which halny wind occurred (Table 6).

### 3.6. Assessment of the Association between Environmental Variables and Number of Cardiac Events

The analyses did not confirm the regression model which assumed that the different environmental variables would impact the number of all cardiac events observed on all the days of the halny months (*p* = 0.866, F = 0.37). This regression model was only explained at a level of 11.0% (Table 7).

Likewise, the analyses did not confirm the regression model for a shorter period of time (multiple regression analysis for 14 days of December), where it was assumed that different environmental variables would be related to the number of cardiac events observed (*p* = 0.451, F = 1.05). This regression model was explained at a level of 39.0% and adjusted at a level of 2.0% (Table 8).

### 3.7. Time Series Data Showing Daily Levels of Environmental Variables and Daily Number of Cardiac Events in the Period from 1 to 14 December 2011

Table 9 presents distributions of mean daily air temperature, atmospheric humidity, speed of wind, pressure, total daily precipitation and daily number of cardiac events, in the period from 1 to 14 December.

Changes in mean daily air temperature, atmospheric humidity, speed of wind, pressure and total daily precipitation and number of cardiac events were shown using linear graphs for the period from 1 to 14 December 2011 (Figure 1 and Figure 2).

Analysis of the environmental variables (i.e., air temperature, atmospheric humidity, speed of wind, pressure and total daily precipitation) failed to demonstrate that any of these variables, or all of them jointly, affected the incidence of cardiac events on the days with halny or on the days preceding its onset.

### 3.8. Assessment of the Number of Admissions Relative to the Environmental Variables, Throughout the Year in the Consecutive Months from January 2011 to December 2011

The relationship of the number of cardiac events to the environmental variables co-occurring in the consecutive months throughout 2011 was examined in Table 10. No significant relationships between the number of cardiac events and the environmental factors were identified in any of the months (whether with or without halny wind) (*p* > 0.05).

## 4. Discussion

Based on a review of the literature it can be concluded that many researchers are investigating the association between wind-related environmental variables, and cardio- or cerebrovascular events, including the impact observed on days preceding onset of such weather conditions [14,15,16,17,18,19,20,21,22,23].

The current study was an attempt to collect evidence related to the impact of foehn wind and the related environmental variables on the incidence of cardiac events. The basic meteotropic factor adopted in the study was defined as halny wind and the accompanying change from cold to warm weather front.

Contrary to the findings reported by other researchers [14,15,16,17,18,19,20,21,22,23], the current study did not confirm effects of environmental factors investigated, i.e., air temperature, atmospheric humidity, speed of wind, pressure and total daily precipitation in the incidence of cardiac events, either on halny days or on days preceding the onset of halny, as well as throughout the whole year in the consecutive months from January 2011 to December 2011. The findings showed no statistically significant correlations between the environmental variables and the number of cardiac events in any of the periods analysed (i.e., days with halny, days preceding halny onset, total days with and preceding halny, other days, all the days combined.) This may be linked with the fact that the data acquired in the current study may be treated as preliminary, and in fact we are planning to carry out further study where we want to acquire more comprehensive data from other hospitals. However, it should be remembered that some of the hospitals may be located further away than others from the Tatra Mountains. Hence, it may be difficult to compare the acquired results. Likewise, Honda et al. explained that their study focused on the association between meteorological factors and frequency of acute myocardial infarction in patients residing in a small, strictly defined region, because the potential variations in weather conditions that are likely to occur in a larger area would have adversely affected their analyses [17]. In view of the above, in our study we have collected evidence on the effects of halny wind in this particular territory. Nevertheless, it is necessary to conduct further research which will take into account other factors, such as air pollution, individuals’ socioeconomic status, lifestyle, body mass index, physical activity, etc. because, based on our findings it may be hypothesised that since there are no correlations between the environmental variables and the number of cardiac events, incidence of myocardial infarction may be linked with other factors rather than weather. These issues will be investigated in our future studies. Other researchers also point out that it is necessary to assess other risk factors that may influence the relationships of environmental variables and the incidence of cardiac events [14,20,21,22].

Although the findings covering the relevant period of time do not show significantly higher number of cardiac events on halny days, a greater number of the related hospital admissions was identified in the case of days immediately preceding occurrence of halny. On the days preceding halny wind onset, the incidence of cardiac events was higher than on the remaining days. However, the findings showed no statistically significant correlation between environmental variables and the number of cardiac events observed during this time. Halny wind frequently leads to a considerable increase in the temperature and decrease in humidity on the leeward side of the mountains. The strong abrupt gusts of wind at times reach a speed typical of a hurricane [4]. Despite the fact that no significant correlations were observed between environmental variables and cardiac events on days immediately preceding halny onset, the graphs in Figure 1 and Figure 2 clearly show that on the consecutive days from 1 to 14 December, which included seven days with halny, the mean temperature in most cases was lower on the day preceding halny, compared to the day with halny onset; atmospheric humidity in most cases was lower on the day preceding halny compared to the day with halny onset; similarly speed of wind on the days preceding halny in most cases was higher than or equal to the speed of wind on the day with halny onset. Hence, it appears that the weather conditions observed on the days preceding halny onset may exacerbate the risk of myocardial infarction, a conclusion which is consistent with opinions presented by other researchers. For example, Honda et al. found that increased incidence of acute myocardial infarction corresponds to lower minimum temperature occurring on the second day preceding the onset of wind [17]. Similarly, Vencloviene et al. showed that prognostic factors for the risk of acute coronary syndromes include such environmental variables as wind speed, daily air temperature, and barometric pressure on the day of the onset and on the two days preceding the ambulance call [18]. Furthermore, Delyukov et al. and Głuszak et al. suggest that similar synchronicities can be observed during a period ranging from 3 to 24 h prior to an arrival of a weather front [30,31]. Analysis of the mean pressure and the number of cardiac events in the consecutive days from 1 to 14 December, including seven days with halny, shows that in most cases pressure was higher on the days preceding halny, compared to the days with halny onset, and the number of cardiac events was higher on the days preceding halny in six cases and only in one case was lower than on the day of halny onset. It can be speculated that the greater number of cardiac incidents identified in our study 24 h before onset of halny wind may be associated with the fact that indeed the wind is only an effect of earlier turbulent changes in atmospheric pressure systems. The factor of key importance here may be the drop in atmospheric pressure, since onset of halny is determined by co-occurrence of high pressure on the southern side and low pressure on the northern side of the Tatra mountains. Other published studies confirm that changes in atmospheric pressure produce adverse effects in the human body, possibly also leading to cardiac incidents [33,34,35,36]. However, our study did not demonstrate statistically significant correlations between cardiac events and pressure, which may result from the short period of the observations. Therefore, in order to establish a cause-and-effect relationship and some temporal patterns, it is necessary to conduct a more comprehensive study taking into account more data from a larger number of hospitals and related to a longer period of time. By expanding the research to include additional years of data it may be possible to increase these numbers and further determine what role temporal effects play.

## 5. Strengths and Limitations of This Study

This is the first exploratory article, the aim of which is to provoke discussion on the impact of foehn wind and the related environmental variables on the incidence of cardiac events in the population inhabiting a region affected by halny wind in Poland. The first study limitation lies in the fact that although the obtained data relate to 465 cardiology patients and cover a 365 day period, in fact the halny days and number of admissions on these days or days either side of halny days only account for a small portion of the total data. To establish a cause-and-effect relationship the relevant data should be acquired from a larger area affected by halny wind in Poland. The second limitation of the study is linked with the insufficient duration of the observation period, and the fact that the analyses took into account data related to one calendar year only; additional years of data would be helpful to increase these numbers and further determine what role temporal effects play. The third limitation results from the fact that we did not consider certain risk factors, such as air pollution and individuals’ socioeconomic status, lifestyle, body mass index and physical activity, that may influence the relationships of environmental variables on the incidence of cardiac events.

## 6. Conclusions

The study failed to demonstrate statistically significant correlations between the environmental factors and the number of cardiac indicants on the days with halny, as well as days preceding halny onset and the remaining days.On the days with halny wind, incidence of cardiac events is similar to that on the remaining days of the year.Overall, we believe that our preliminary results may contribute to the discussion concerning the impact of foehn wind and associated environmental factors on the incidence of myocardial infarction in people living in the areas affected by halny wind in Poland. This analysis suggests that in order to establish a cause-and-effect relationship, more comprehensive data should be acquired from a larger number of regions affected by halny wind in Poland, and the relevant observations should be continued for a period longer than one calendar year. It will also be necessary to take into account effects produced by other factors, such as air pollution, individuals’ socioeconomic status, lifestyles, body mass index and physical activity.

## Figures and Tables

**Figure 1 ijerph-17-02638-f001:**
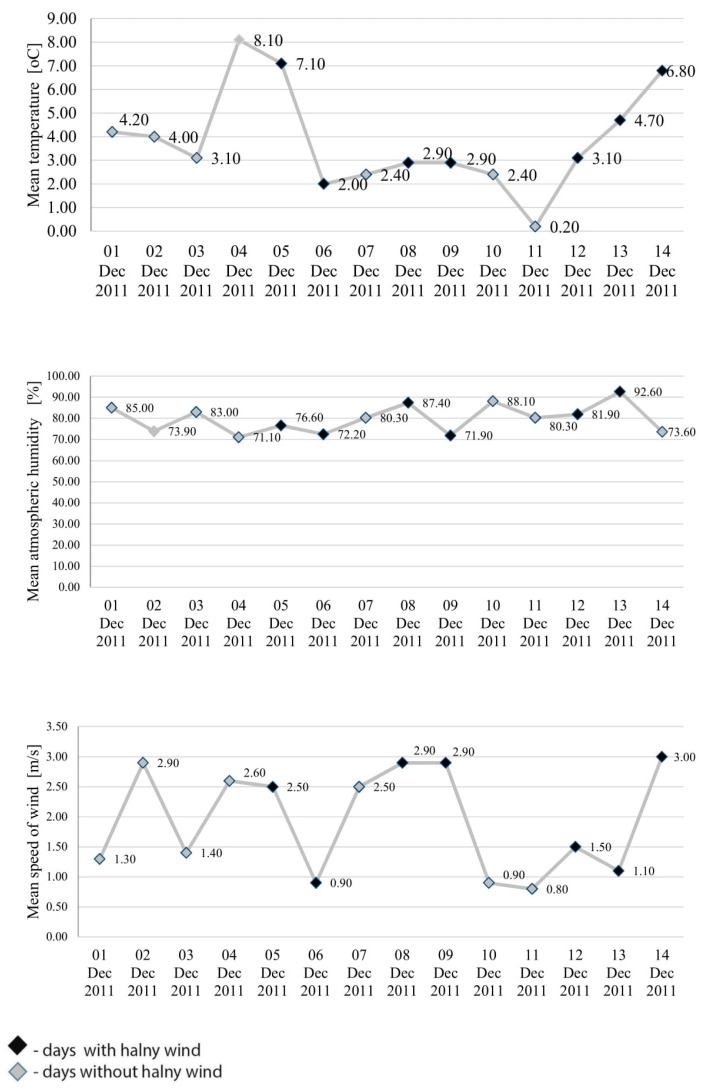
Changes in mean daily air temperature, atmospheric humidity and speed of wind for the period from 1 to 14 December 2011.

**Figure 2 ijerph-17-02638-f002:**
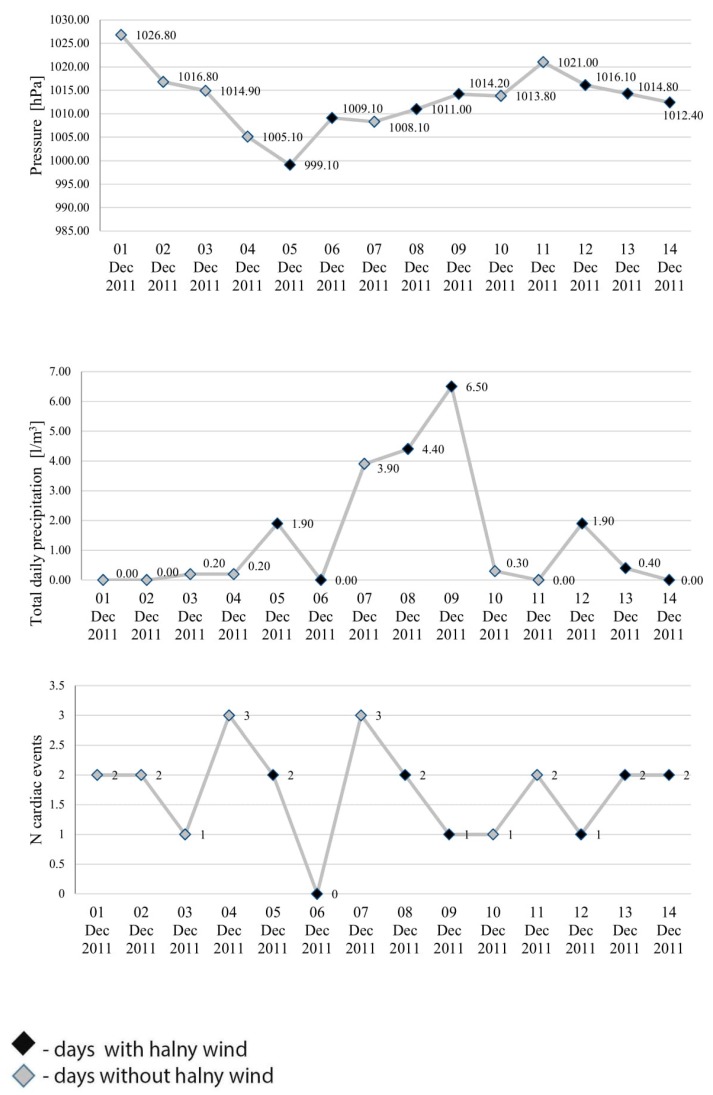
Changes in mean pressure, total daily precipitation and number of cardiac events for the period from 1 to 14 December 2011.

**Table 1 ijerph-17-02638-t001:** Characteristics of the study group.

Number	Females	Males
Observed (O)	168	297
% of the total number	36.1%	63.9%
Expected €	232.5	232.5
% of the total number	50.0%	50.0%
Significance level—p	<0.001	

**Table 2 ijerph-17-02638-t002:** Occurrence of halny and number of days affected in year 2011.

Month	Dates	Number of Halny Days
January	6–7 Jan, 26 Jan	3
March	12–14 March, 17 March	4
October	25–26 Oct	2
November	4–6 Nov	3
December	5–6 Dec, 8–9 Dec, 12–17 Dec	10
		TOTAL 22

**Table 3 ijerph-17-02638-t003:** Admissions in months with halny wind. Number of admissions on halny days vs. remaining days.

Sex	Days with Halny Wind	Days without Halny
Total admissions	25 (11%)	202 (89%)
Females admitted	10 (13.3%)	66 (86.8%)
Males admitted	15 (9.9%)	136 (90.1%)
Chi-square	0.54	
Degrees of freedom—df	1	
Significance level—p	0.46	

**Table 4 ijerph-17-02638-t004:** Number of admissions on halny days and on the remaining days of the year.

Days	Number of Admissions	Number of Days
Halny days	25	22
Remaining days	440	343
Total	465	365
	Significance level—*p*	0.496

*p*—level of probability in the test for significance of structure indicators.

**Table 5 ijerph-17-02638-t005:** Number of admissions on halny days, the remaining days during halny months and one day before onset of halny, and on the remaining days of months with halny.

Days	Number of Admissions	Number of Days
Halny days	25	22
Remaining days	202	132
Total	227	154
	Significance level—*p*	0.084
Days immediately preceding onset of halny	18	8
Remaining days	209	146
Total	227	154
	Significance level—*p*	0.001

*p*—level of probability in the test for significance of structure indicators.

**Table 6 ijerph-17-02638-t006:** Relationship between the number of cardiac events and environmental variables in the selected periods of time.

Environmental Variables	Halny Days	Days Preceding Halny	Total Number of Days with Halny and Preceding Halny	Other Days	Total Number of Days
Mean air temperature (°C)	0.33	0.21	0.11	0.02	0.09
Mean atmospheric humidity (%)	*p* = 0.140	*p* = 0.354	*p* = 0.569	*p* = 0.839	*p* = 0.272
Mean speed of wind (m/s)	−0.19	−0.16	0.00	0.15	0.01
Pressure (hPa)	*p* = 0.395	*p* = 0.480	*p* = 0.987	*p* = 0.106	*p* = 0.922
Total daily precipitation (L/m^3^)	−0.17	0.04	−0.16	−0.03	−0.039
Mean air temperature (°C)	*p* = 0.461	*p* = 0.855	*p* = 0.392	*p* = 0.746	*p* = 0.631
Mean atmospheric humidity (%)	0.21	0.34	0.30	0.06	−0.08
Mean speed of wind (m/s)	*p* = 0.353	*p* = 0.116	*p* = 0.106	*p* = 0.528	*p* = 0.346
Pressure (hPa)	−0.29	−0.37	−0.30	0.09	0.04
	*p* = 0.195	*p* = 0.093	*p* = 0.098	*p* = 0.340	*p* = 0.633

*p*—level of probability for significance of Pearson’s correlation coefficient.

**Table 7 ijerph-17-02638-t007:** Assessment of the number of cardiac events/number of admissions relative to the environmental variables on all days of the halny months.

Environmental Variables vs. Number of Cardiac Events	Multiple Regression						
	R^2^	Adjusted R^2^	F	*p* for Regression Model	b	Partial Correlation	*p*
Mean air temperature (°C)	0.11	0.01	0.37	0.866	0.01	0.04	0.586
Mean atmospheric humidity (%)					−0.00	−0.01	0.873
Mean speed of wind (m/s)					−0.00	−0.04	0.613
Pressure (hPa)					−0.01	−0.07	0.413
Total daily precipitation (L/m^3^)					0.00	0.00	0.962

R^2^—regression model. Adjusted R^2^—regression model disregards extreme values. F—result of Fisher’s test. b—regression coefficient. Partial correlation—the variable Xi is correlated with the variable Y after effect of all the other independent variables is taken into account (concurrent effect of all the four factors). *p*—significance level.

**Table 8 ijerph-17-02638-t008:** Assessment of the number of cardiac events/number of admissions relative to the environmental variables from 1 to 14 December.

Environmental Variables vs. Number of Cardiac Events	Multiple Regression						
	R^2^	Adjusted R^2^	F	*p* for Regression Model	b	Partial Correlation	*p*
Mean air temperature (°C)	0.39	0.02	1.05	0.451	0.06	0.11	0.753
Mean atmospheric humidity (%)					0.05	0.38	0.283
Mean speed of wind (m/s)					0.68	0.45	0.197
Pressure (hPa)					0.00	0.01	0,970
Total daily precipitation (L/m^3^)					−0.11	−0.22	0.546

R^2^—regression model. Adjusted R^2^—regression model disregards extreme values. F—result of Fisher’s test. b—regression coefficient. Partial correlation—the variable Xi is correlated with the variable Y after effect of all the other independent variables is taken into account (concurrent effect of all the four factors). *p*—significance level.

**Table 9 ijerph-17-02638-t009:** Example rows of time series data from the Tarnów dataset showing daily levels of environmental variables and daily number of cardiac events.

Date	Mean Air Temperature (°C)	Mean Atmospheric Humidity (%)	Mean Speed of Wind (m/s)	Pressure (hPa)	Total Daily Precipitation (L/m^3^)	N Cardiac Events
1 Dec. 11	4.20	85.00	1.30	1026.80	0.00	2
2 Dec. 11	4.00	73.90	2.90	1016.80	0.00	2
3 Dec. 11	3.10	83.00	1.40	1014.90	0.20	1
4 Dec. 11	8.10	71.10	2.60	1005.10	0.20	3
5 Dec. 11	7.10	76.60	2.50	999.10	1.90	2
6 Dec. 11	2.00	72.50	0,90	1009.10	0.00	0
7 Dec. 11	2.40	80.30	2.50	1008.30	3.90	3
8 Dec. 11	2.90	87.40	2.90	1011.00	4.40	2
9 Dec. 11	2.90	71.90	2.90	1014.20	6.50	1
10 Dec. 11	2.40	88.10	0.90	1013.80	0.30	1
11 Dec. 11	0.20	80.30	0.80	1021.00	0.00	2
12 Dec. 11	3.10	81.90	1.50	1016.10	1.90	1
13 Dec. 11	4.70	92.60	1.10	1014.30	0.40	2
14 Dec. 11	6.80	73.60	3.00	1012.40	0.00	2

**Table 10 ijerph-17-02638-t010:** Assessment of the number of cardiac events/number of admissions relative to the environmental variables from January 2011 to December 2011.

Month	Environmental Variables vs Number of Cardiac Events			
	R^2^	Adjusted R^2^	F	*p*
January *	0.31	0.09	0.52	0.760
February	0.51	0.26	1.52	0.226
March *	0.46	0.21	1.36	0.274
April	0.41	0.17	1.04	0.415
May	0.54	0.29	2.04	0.107
June	0.21	0.05	0.23	0.945
July	0.39	0.15	0.93	0.476
August	0.50	0.25	1.64	0.187
September	0.44	0.19	1.15	0.360
October *	0.38	0.14	0.84	0.533
November *	0.29	0.08	0.44	0.818
December *	0.33	0.11	0.63	0.679

R^2^—regression model. Adjusted R^2^—regression model disregards extreme values. F—result of Fisher’s test. *p*—significance level. * months with halny.

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
