# Peer review of "Impact of Foehn Wind and Related Environmental Variables on the Incidence of Cardiac Events"

_ijerph, 2020, doi:10.3390/ijerph17082638_

Round 1

Reviewer 1 Report

Background (and/or discussion) could be improved by incorporating selected references.

1) Importance of abrupt changes in meteorological variables (most cited examples):

Significant 6-month periodicity in the onset of subarachnoid haemorrhage (SAH), with two peaks in spring and autumn. Importance of the transitional seasons, when frequent stress episodes exerted on the arterial wall by typical temperature excursions may precipitate aneurysmal rupture to a greater degree than the more stable (but not rigid) winter cold.

Gallerani M, et al. Circadian and circannual rhythmicity in the occurrence of subarachnoid hemorrhage. Stroke. 1996;27(10):1793-7.

International Worldwide Registry. The winter peak of acute aortic dissection (AAD) is evident in both cold and temperate climate settings, suggesting that the relative change in temperature, rather than absolute temperature, and/or endogenous annual rhythms are critical mechanistic factors.

Mehta RH, et al; International Registry of Acute Aortic Dissection (IRAD) Investigators. The winter peak in the occurrence of acute aortic dissection is independent of climate. Chronobiol Int. 2005;22(4):723-9. 

2) I would suggest to discuss (or incorporate, for example in a table) some previous recent findings on the association between wind and cardio- and cerebrovascular events, and the impact even on the preceding days. A brief list of possible topic and papers are enclosed.

Acute cardiovascular diseases

Acute cardiovascular diseases (ACVDs): positive significant association between ACVDs and a cold front effect lagged by one day (p=0.018) with a relative risk (RR) of 1.095.

Boussoussou M, et al. Atmospheric fronts as minor cardiovascular risk factors, a new approach to preventive cardiology. J Cardiol. 2020;75(2):196-202.

Myocardial infarction (MI)

Incidence of MI increased with lower air temperature, lower atmospheric air pressure, higher wind velocity, and shorter sunshine duration.

Mohammad MA, et al. Association of weather with day-to-day incidence of myocardial infarction: A SWEDEHEART Nationwide Observational Study. JAMA Cardiol. 2018;3(11):1081-19.

Lower minimum temperature on the 2nd day preceding the onset is an independent risk factor for the frequent onset of AMI.

Honda T, et al. Influence of weather conditions on the frequent onset of acute myocardial infarction. J Cardiol. 2016;67(1):42-50.

Weak inverse correlations between atmospheric air temperatures, rainfall level and direct correlation between air wind speed, atmospheric pressure and MI morbidity were established.

Radišauskas R, et al. Trends of myocardial infarction morbidity and its associations with weather conditions. Medicina (Kaunas). 2014;50(3):182-9.

Acute coronary events (emergency calls)

Daily air temperature, barometric pressure, and wind speed (WS) on the day of the ambulance call and on the two previous days may be prognostic variables for the risk of acute coronary syndromes.

Vencloviene J, et al. Effects of weather conditions on emergency ambulance calls for acute coronary syndromes. Int J Biometeorol. 2015;59(8):1083-93.

Associations between changes in weather conditions (including an excess number of wind speed increases) and the likelihood of an ACS event.

Ezekowitz JA, et al. The relationship between meteorological conditions and index acute coronary events in a global clinical trial. Int J Cardiol. 2013;168(3):2315-21.

Out-of-hospital cardiac arrest of presumed cardiac etiology (OHCA)

Probability of OHCA-occurrence increased continuously with increasing wind speed (p < 0.001). There are associations between several weather-related factors such as temperature, humidity, air pressure, and wind speed, and occurrence of OHCA of presumed cardiac etiology.

Hensel M, et al. Association between weather-related factors and cardiac arrest of presumed cardiac etiology: A prospective observational study based on out-of-hospital care data. Prehosp Emerg Care. 2018;22(3):345-52.

Paroxysmal atrial fibrillation (emergency ambulance calls (EC)

At night, the elevated risk was associated with wind speed above the median (lag 2-4 days) (per 1-kt increase: RR = 1.07)

Vencloviene J, et al. The short-term associations of weather and air pollution with emergency ambulance calls for paroxysmal atrial fibrillation. Environ Sci Pollut Res Int. 2017;24(17):15031-43.

Acute cerebrovascular events

High wind speed on the event day associated with higher incidence of ischemic stroke in older subjects.

Tamasauskiene L, et al. Relationship of meteorological factors and acute stroke events in Kaunas (Lithuania) in 2000-2010. Environ Sci Pollut Res Int. 2017;24(10):9286-93.

The ORs for wind speed, daily wind speed range (DWR), and wind chill for ischemic stroke patients were slightly greater than for patients in the total sample (OR=1.20 and OR=1.22, respectively).

Kim J, et al. The association between wind-related variables and stroke symptom onset: A case-crossover study on Jeju Island. Environ Res. 2016;150:97-105.

Author Response

Response to Reviewer 1 Comments

Dear Reviewer,        

We thank you for reviewing our article titled, “Impact of foehn wind and related environmental variables on the incidence of cardiac events”.  We have made every effort to improve our manuscript, as guided by the reviewer’s helpful suggestions.

We thank the reviewer of all the comments. Answers are summarised below. All changes are highlighted as red text in the manuscript.

We hope you will be pleased with the changes, and support the publication of our revised manuscript.

With kind regards,

The authors of the article

Point 1: Background (and/or discussion) could be improved by incorporating selected references.

1) Importance of abrupt changes in meteorological variables (most cited examples):

Significant 6-month periodicity in the onset of subarachnoid haemorrhage (SAH), with two peaks in spring and autumn. Importance of the transitional seasons, when frequent stress episodes exerted on the arterial wall by typical temperature excursions may precipitate aneurysmal rupture to a greater degree than the more stable (but not rigid) winter cold.

Gallerani M, et al. Circadian and circannual rhythmicity in the occurrence of subarachnoid hemorrhage. Stroke. 1996;27(10):1793-7.

International Worldwide Registry. The winter peak of acute aortic dissection (AAD) is evident in both cold and temperate climate settings, suggesting that the relative change in temperature, rather than absolute temperature, and/or endogenous annual rhythms are critical mechanistic factors.

Mehta RH, et al; International Registry of Acute Aortic Dissection (IRAD) Investigators. The winter peak in the occurrence of acute aortic dissection is independent of climate. Chronobiol Int. 2005;22(4):723-9. 

2) I would suggest to discuss (or incorporate, for example in a table) some previous recent findings on the association between wind and cardio- and cerebrovascular events, and the impact even on the preceding days. A brief list of possible topic and papers are enclosed.

Acute cardiovascular diseases

Acute cardiovascular diseases (ACVDs): positive significant association between ACVDs and a cold front effect lagged by one day (p=0.018) with a relative risk (RR) of 1.095.

Boussoussou M, et al. Atmospheric fronts as minor cardiovascular risk factors, a new approach to preventive cardiology. J Cardiol. 2020;75(2):196-202.

Myocardial infarction (MI)

Incidence of MI increased with lower air temperature, lower atmospheric air pressure, higher wind velocity, and shorter sunshine duration.

Mohammad MA, et al. Association of weather with day-to-day incidence of myocardial infarction: A SWEDEHEART Nationwide Observational Study. JAMA Cardiol. 2018;3(11):1081-19.

Lower minimum temperature on the 2nd day preceding the onset is an independent risk factor for the frequent onset of AMI.

Honda T, et al. Influence of weather conditions on the frequent onset of acute myocardial infarction. J Cardiol. 2016;67(1):42-50.

Weak inverse correlations between atmospheric air temperatures, rainfall level and direct correlation between air wind speed, atmospheric pressure and MI morbidity were established.

Radišauskas R, et al. Trends of myocardial infarction morbidity and its associations with weather conditions. Medicina (Kaunas). 2014;50(3):182-9.

Acute coronary events (emergency calls)

Daily air temperature, barometric pressure, and wind speed (WS) on the day of the ambulance call and on the two previous days may be prognostic variables for the risk of acute coronary syndromes.

Vencloviene J, et al. Effects of weather conditions on emergency ambulance calls for acute coronary syndromes. Int J Biometeorol. 2015;59(8):1083-93.

Associations between changes in weather conditions (including an excess number of wind speed increases) and the likelihood of an ACS event.

Ezekowitz JA, et al. The relationship between meteorological conditions and index acute coronary events in a global clinical trial. Int J Cardiol. 2013;168(3):2315-21.

Out-of-hospital cardiac arrest of presumed cardiac etiology (OHCA)

Probability of OHCA-occurrence increased continuously with increasing wind speed (p < 0.001). There are associations between several weather-related factors such as temperature, humidity, air pressure, and wind speed, and occurrence of OHCA of presumed cardiac etiology.

Hensel M, et al. Association between weather-related factors and cardiac arrest of presumed cardiac etiology: A prospective observational study based on out-of-hospital care data. Prehosp Emerg Care. 2018;22(3):345-52.

Paroxysmal atrial fibrillation (emergency ambulance calls (EC)

At night, the elevated risk was associated with wind speed above the median (lag 2-4 days) (per 1-kt increase: RR = 1.07)

Vencloviene J, et al. The short-term associations of weather and air pollution with emergency ambulance calls for paroxysmal atrial fibrillation. Environ Sci Pollut Res Int. 2017;24(17):15031-43.

Acute cerebrovascular events

High wind speed on the event day associated with higher incidence of ischemic stroke in older subjects.

Tamasauskiene L, et al. Relationship of meteorological factors and acute stroke events in Kaunas (Lithuania) in 2000-2010. Environ Sci Pollut Res Int. 2017;24(10):9286-93.

The ORs for wind speed, daily wind speed range (DWR), and wind chill for ischemic stroke patients were slightly greater than for patients in the total sample (OR=1.20 and OR=1.22, respectively).

Kim J, et al. The association between wind-related variables and stroke symptom onset: A case-crossover study on Jeju Island. Environ Res. 2016;150:97-105.

Response 1: Thank you for this valuable comment. We fully agree with the Reviewer. In accordance with the Reviewer’s suggestion, we have improved Background and Discussion by incorporating these references. We have discussed recent findings on the association between wind and cardio- and cerebrovascular events, and the impact also observed on the preceding days.

BEFORE

lines 32-71: 1. Introduction

Cardiovascular disorders are the most common cause of mortality worldwide, including most European countries, and Poland [1]. Seasonal peaks in acute myocardial infarction incidence have been widely reported. Weather has been postulated to be one of the elements greatly contributing to this association [2]. Foehn wind may be one of the weather-related factors which may provoke cardiac incidents [3-8].

The term “föhn” is used with reference to a type of strong and turbulent wind blowing down from mountain crests towards the lee side, and bringing dry and warm air to the surface. Foehn events rather frequently occur in the Alps, or more specifically on the northern side of the mountains, and constitute a significant challenge in weather forecasting for the Alpine valleys, inhabited by a lot of people. The northern Alpine foehn is frequently observed in the prefrontal environment. The foehn clearance reflects the leeward range of the foehn in the mountains, i.e. a region with no clouds where dry and warm air occurs at the surface, just as in the foehn region but with no strong winds. This meteorological phenomenon is not unique to the Alps, and it can be observed over other high mountains worldwide [7].

A local variety of foehn, “halny” wind is produced by violent flow of air masses from the southern side of the Carpathians or the Sudetes. This happens when Poland is approached from the west by a deep cyclone (low-pressure area), while the centre of anticyclone (high-pressure) is in Ukraine [6,7]. In addition to an abrupt change in temperature and large speed of the wind, there are also other consequences of halny wind. The first one involves rapid pressure drop, and the other one is linked with increased condensation of positive ions in the air as a result of which microscopic particles of gases and aerosols, including harmful elements and chemical compounds, can easily penetrate organisms. This may lead to headaches, muscle and join pain, increased problems related to the cardiovascular and nervous systems, and most commonly to poor mood [6-8].

In the literature we can find reports discussing health-related changes in response to foehn wind. It is believed that foehn wind may also lead to increased aggressiveness, more severe depressive states, emotional crises ending with suicide attempts, agitation, anxiety, fatigue, weariness, problems with concentration, sleep disorders, migraine, muscle and joint pain as well as cardiovascular incidents [3, 4, 6, 8].  The foehn wind known as “halny”, occurring in southern Poland, has equally bad reputation as the foehn winds blowing in other parts of Europe and the world. We decided to investigate its impact on the incidence of cardiac events in the population of southern Poland, a region affected by this type of wind. The present study was encouraged by findings reported in biometeorological literature showing negative effects of other foehn winds, such as "Chinook" descending the slopes of the Rocky Mountains, "Santa Ana" and "Santa Maria" blowing in California, "Megas" in Greece, and "Zonda" blowing in the Andes. Negative effect of this type of wind is sometimes considered as a mitigating circumstance in judicial decisions related to minor offences, e.g. in Switzerland and in the Mediterranean countries affected by sirocco and chamsin winds [4,5].

The aim of this study was to examine the impact of foehn wind on the incidence of cardiac events in the population inhabiting a region affected by halny wind.

lines 140-212: 4. Discussion

People with high sensitivity to weather changes are referred to as meteoropaths. They present increased tension of the vegetative system and poor adaptative responses. Effects of weather fronts, in particular cold fronts, include decrease in the temperature and increase in the atmospheric pressure, responses to which include vasoconstriction, potentially posing a burden for the circulatory system [11-13] Nowadays, it is estimated that the number of me-teoropaths is consistently growing. They constitute approximately 50%-70% of the healthy populations. It is also believed that meteoropathy is a “disease” of affluence, caused by stress, work overload, and unhealthy lifestyles, since meteoropaths mainly include residents of urban areas, whose immunity is affected by the long time spent in the artificial microcli-mates of their homes and offices, by a lack of exercise and by polluted air. It is also pointed out that meteoropathy is most commonly observed in individuals affected by coronary dis-ease, hypertension and ulcer disease [11-14].

Based on a review of the literature it can be concluded that in many geographical regions of the world a relationship was found between weather conditions, including foehn winds, and frequency of hospital admission due to myocardial infarction [15-20]. However, so far in Poland there is no data related to the impact of foehn wind on the incidence of cardiac events.  This observation provided motivation for the current study.

Clinical studies provide evidence that some meteorological factors, and most important-ly weather changes occurring rapidly, e.g. in one day, may play an important role in induc-ing subjectively perceived indisposition in healthy individuals and in promoting disease symptoms in patients. Notably, the same weather changes may induce completely different bodily responses. As an example, a large drop in barometric pressure will induce a decrease in blood pressure in some individuals and an increase in other people [2-5, 21-24]. Due to this no uniform recommendations for dealing with sudden changes in weather can be ad-dressed e.g. to all patients with cardiovascular problems.

The current study was an attempt to collect evidence related to the impact of foehn wind on the incidence of cardiac events. The basic meteotropic factor adopted in the study was defined as halny wind and the accompanying change from cold to warm weather front. The acquired results partly provide evidence related to effects of halny wind in the incidence of acute myocardial infarction. Although the findings covering the relevant period of time do not show significantly higher number of cardiac events on halny days, such correlation was identified in the case of days immediately preceding occurrence of halny. On days preceding halny wind onset, incidence of cardiac events was higher than on the remaining days. Re-search conducted by other authors also suggests occurrence of similar synchronicities during a period ranging from 3 to 24 hours prior to an arrival of a weather front [27,28]. It has also been shown that meteorotropic responses in both healthy people and individuals affected by medical conditions may be observed more in advance of specific changes in weather; in ex-treme cases these may be “sensed” up to 48 hours before they happen. Such sensitivity may result from changes in electric field of the atmosphere, preceding arrival of a weather front [29]. It can be speculated that the greater number of cardiac incidents identified in our study 24 hours before onset of halny wind may be associated with the fact that indeed the wind is only an effect of earlier turbulent changes in atmospheric pressure systems. Hence, it is not halny wind itself that impacts cardiac events but the preceding changes in weather fronts. The factor of key importance here may be the drop in atmospheric pressure since onset of halny is determined by co-occurrence of high pressure on the southern side and low pressure on the northern side of the Tatra mountains. And this pressure drop may impact cardiac events. Other published studies confirm that changes in atmospheric pressure produce ad-verse effects in human body, possibly also leading to cardiac incidents [30-33].

As early as 1953, it was suggested by Petersen that human organism may be stimulated by specific relationships between weather fronts [4]. According to data from the Institute of Meteorology and Water Management, as many as 50-70% of Poles respond to weather changes [25]. Cardiovascular problems are most frequently listed among meteorotropic dis-eases. In Poland the earliest studies focusing on impact of weather on functioning of cardio-vascular system were carried out in the mid-1800s [4]. Research shows impact of short-term weather changes on intensity of ischemic heart disease symptoms, increased oscillations in arterial blood pressure, and higher incidence of acute myocardial infarction. Over 34% of fatal cases of myocardial infarction co-occur with a passage of weather fronts, compared to only 6% observed during periods of calm weather linked with high pressure [26]. A survey conducted by Baranowska showed that in the respondents’ opinion, meteoropaths are ad-versely affected by weather transitions from highs to lows [4]. In a study conducted by Höppe, individuals sensitive to weather, living in northern Germany, negatively assessed stormy weather (30%) and influx of cold air (29%). On the other hand, inhabitants of south-ern Germany experience mood decline accompanying influx of warm (30%) rather than cold air (18%). This difference may be associated with the quite frequent foehn winds in southern Germany [16]. Mackensen et al. investigated effects of weather in the sense of well-being observed in people living in Germany and in Canada. The authors reported weather sensitivi-ty in 54.5% of the subjects representing German population while in Canada the rate amounted to 61% [17]. Likewise, Shiue et al. found a relationship between weather condi-tions, including wind speed, and hospital admissions due to heart disease in Germany [18]. The same relationship was shown by a study conducted in Lithuania [20].

It is difficult to clearly establish associations between meteorological or synoptic factors and decline in people’s mood or well-being. The reasons for that may lie e.g. in var-ied responses to weather related stimuli, or in synergistic effects produced by various mete-orological factors. Therefore, results obtained by various researchers frequently cannot be compared due to the fact that different methodologies are applied in studies.

AFTER

lines 33-96: 1. Introduction

Cardiovascular disorders are the most common cause of mortality worldwide, including most European countries, and Poland [1]. Seasonal peaks in acute myocardial infarction incidence have been widely reported. Weather has been postulated to be one of the elements greatly contributing to this association [2].

The term “föhn” is used with reference to a type of strong and turbulent wind blowing down from mountain crests towards the lee side, and bringing dry and warm air to the surface. Foehn events rather frequently occur in the Alps, or more specifically on the northern side of the mountains, and constitute a significant challenge in weather forecasting for the Alpine valleys, inhabited by a lot of people. The northern Alpine foehn is frequently observed in the prefrontal environment. The foehn clearance reflects the leeward range of the foehn in the mountains, i.e. a region with no clouds where dry and warm air occurs at the surface, just as in the foehn region but with no strong winds. This meteorological phenomenon is not unique to the Alps, and it can be observed over other high mountains worldwide [3].

A local variety of foehn, “halny” wind is produced by violent flow of air masses from the southern side of the Carpathians or the Sudetes. This happens when Poland is approached from the west by a deep cyclone (low-pressure area), while the centre of anticyclone (high-pressure) is in Ukraine [3,4]. In addition to an abrupt change in temperature and large speed of the wind, there are also other consequences of halny wind. The first one involves rapid pressure drop, and the other one is linked with increased condensation of positive ions in the air as a result of which microscopic particles of gases and aerosols, including harmful elements and chemical compounds, can easily penetrate organisms [3,4]. Halny wind frequently leads to considerable increase in the temperature and decrease in humidity on the leeward side of the mountains. The strong abrupt gusts of wind at times reach a speed typical for hurricane, and are accompanied with characteristic calm intervals. The foehn winds are observed in the Tatras rather frequently, particularly during the cold seasons. The criteria commonly applied by Polish researchers investigating foehn winds occurring in the mountainous areas include: direction (usually 90-270) and wind speed (> 4 m/s, commonly ≥10 m/s, at times increasing to even 60 m/s), occurrence of strong gusts and calm intervals, sudden rise in the air temperature (by several degrees in a short time) and decrease in the humidity (commonly ≤70%), changes in atmospheric pressure and / or certain types of the clouds. In foothill and lowland areas, the wind is dry and warm [4].

It is believed that foehn wind may lead to increased aggressiveness, more severe depressive states, emotional crises ending with suicide attempts, agitation, anxiety, fatigue, weariness, muscle and joint pain, problems with concentration, sleep disorders, migraine, as well as headaches [4-7]. 

People with high sensitivity to weather changes are referred to as meteoropaths. They present increased tension of the vegetative system and poor adaptative responses. Effects of weather fronts, in particular cold fronts, include decrease in the temperature and increase in the atmospheric pressure, responses to which include vasoconstriction, potentially posing a burden for the circulatory system [8-10] Nowadays, it is estimated that the number of meteoropaths is consistently growing. According to the data from the Institute of Meteorology and Water Management, as many as 50-70% of Poles respond to weather changes [11]. Similar statistics have been published in other countries, e.g. in Germany 54.5% of the population is weather sensitive, in Canada the relevant rate amounts to 61% [12]. It is also believed that meteoropathy is a “disease” of affluence, caused by stress, work overload, and unhealthy lifestyles, since meteoropaths mainly include residents of urban areas, whose immunity is affected by the long time spent in the artificial microclimates of their homes and offices, by a lack of exercise and by polluted air. It has also been pointed out that meteoropathy is most commonly observed in individuals affected by coronary disease, hypertension and ulcer disease [8-10,13].

A large body of evidence reported in the literature shows there is an association between wind- related environmental variables, and cardiovascular or cerebrovascular events, with impact observed also on the preceding days [14-23]. The related research findings focus on acute cardiovascular diseases and atmospheric front effect lagged by one day [14], myocardial infarction in relation to wind velocity, atmospheric air pressure, air temperature [15, 16] and lower minimum temperature on the 2nd day preceding the onset [17], as well as acute coronary events and wind speed [18], and in particular excess number of wind speed increases [19], air temperature, and barometric pressure, on the day of the ambulance call and on the two previous days [18]. Associations have also been reported between out-of-hospital cardiac arrest of presumed cardiac etiology and wind speed, temperature, humidity, air pressure [20], paroxysmal atrial fibrillation and wind speed [21] as well as acute cerebrovascular events and wind speed [22, 23] or wind chill [23].

The foehn wind known as “halny”, occurring in southern Poland, has equally bad reputation as the foehn winds blowing in other parts of Europe and the world. The aim of this study was to examine the impact of the foehn wind and the related environmental variables on the incidence of cardiac events in the population inhabiting a region affected by halny wind. We also decided to determine whether the environmental changes coincide with or predate the event examined?

lines 236-331: 4. Discussion

Based on a review of the literature it can be concluded that many researchers are investigating the association between wind-related environmental variables, and cardio- or cerebrovascular events, including the impact observed on days preceding onset of such weather conditions [14-23]. However, so far in Poland no such studies have been published yet. Boussoussou et al. have reported positive significant association between acute cardiovascular diseases and a cold front effect lagged by one day [14]. Mohammad et al. demonstrated that incidence of myocardial infarction increased with higher wind velocity, lower air temperature, lower atmospheric air pressure, and shorter sunshine duration [15]. Similarly, Honda et al. found that increased incidence of acute myocardial infarction corresponds to lower minimum temperature occurring on the 2nd day preceding the onset of wind [17]. Radišauskas et al. establish that myocardial infarction morbidity is directly correlated to wind speed, and atmospheric pressure and inversely correlated to atmospheric temperatures, and rainfall level [16]. Goerre et al. reported that the incidence of myocardial infarctions is significantly affected by wind activity, ambient pressure, and pressure gradient [15]. Vencloviene et al. showed that prognostic factors for the risk of acute coronary syndromes include such environmental variables as wind speed, daily air temperature, and barometric pressure on the day of the onset and on the two days preceding the ambulance call [18]. Likewise, Ezekowitz et al. showed relationships between the likelihood of an acute coronary event and rapid changes in weather conditions (including excessively frequent increases in wind speed) [19]. Hensel et al. showed that increasing wind speed corresponds to greater likelihood of out-of-hospital cardiac arrest of presumed cardiac aetiology [20]. Vencloviene et al. demonstrated that increased risk of paroxysmal atrial fibrillation occurring at night was linked with wind speed above the median (lag 2-4 days) [21]. Tamasauskiene et al. pointed out that a greater risk of ischemic stroke in older patients was associated with high wind speed on the day of the incident [22]. Similarly, Kim et al. showed that the odds ratios for wind speed, daily wind speed range, and wind chill for patients with ischemic stroke were slightly greater than in the total sample [23].

Effects of seasonal changes in environmental variables have also been pointed out. According to Gallerani et al., it is possible to observe 6-month-period regularities in the incidence of subarachnoid haemorrhage, spring and autumn being the peak seasons. The authors propose an explanation that during these transitional seasons the arterial wall is exposed to frequent stress episodes due to typical changes in temperature and that may result in aneurysmal rupture occurring more frequently than e.g. during the more stable periods of winter cold [26]. Investigators from the International Worldwide Registry reported that the high incidence of acute aortic dissection in winter is observed in various climate settings (both cold and temperate), which seems to suggest that the major mechanistic factors contributing to the phenomenon include the relative change in temperature, rather than absolute temperature, and/or circannual rhythms [27].

Contrary to the findings reported by other researchers [14-23], the current study did not confirm effects of environmental factors investigated, i.e. air temperature, atmospheric humidity, speed of wind, pressure and total daily precipitation in the incidence of cardiac events, either on halny days, or on days preceding the onset of halny.

Clinical studies provide evidence that some meteorological factors, and most importantly weather changes occurring rapidly, e.g. in one day, may play an important role in inducing subjectively perceived indisposition in healthy individuals and in promoting disease symptoms in patients. Notably, the same weather changes may induce completely different bodily responses. As an example, a large drop in barometric pressure will induce a decrease in blood pressure in some individuals and an increase in other people [2, 28-31]. Due to this no uniform recommendations for dealing with sudden changes in weather can be ad-dressed e.g. to all patients with cardiovascular problems.

The current study was an attempt to collect evidence related to the impact of foehn wind and the related environmental variables on the incidence of cardiac events. The basic meteotropic factor adopted in the study was defined as halny wind and the accompanying change from cold to warm weather front. The acquired results partly provide evidence related to effects of halny wind in the incidence of acute myocardial infarction. Although the findings covering the relevant period of time do not show significantly higher number of cardiac events on halny days, such correlation was identified in the case of days immediately preceding occurrence of halny. On the days preceding halny wind onset, incidence of cardiac events was higher than on the remaining days, however the findings showed no effect of the related environmental variables in the number of cardiac events observed during this time. Research conducted by other authors also suggests occurrence of similar synchronicities during a period ranging from 3 to 24 hours prior to an arrival of a weather front [32,33]. It has also been shown that meteorotropic responses in both healthy people and individuals affected by medical conditions may be observed more in advance of specific changes in weather; in extreme cases these may be “sensed” up to 48 hours before they happen. Such sensitivity may result from changes in electric field of the atmosphere, preceding arrival of a weather front [34]. It can be speculated that the greater number of cardiac incidents identified in our study 24 hours before onset of halny wind may be associated with the fact that indeed the wind is only an effect of earlier turbulent changes in atmospheric pressure systems. Hence, it is not halny wind itself that impacts cardiac events but the preceding changes in weather fronts. The factor of key importance here may be the drop in atmospheric pressure since onset of halny is determined by co-occurrence of high pressure on the southern side and low pressure on the northern side of the Tatra mountains. And this pressure drop may impact cardiac events. Other published studies confirm that changes in atmospheric pressure produce ad-verse effects in human body, possibly also leading to cardiac incidents [35-38].

As early as 1953, it was suggested by Petersen that human organism may be stimulated by specific relationships between weather fronts [6]. Cardiovascular problems are most frequently listed among meteorotropic diseases. In Poland the earliest studies focusing on impact of weather on functioning of cardio-vascular system were carried out in the mid-1800s [6]. Research shows impact of short-term weather changes on intensity of ischemic heart disease symptoms, increased oscillations in arterial blood pressure, and higher incidence of acute myocardial infarction. Over 34% of fatal cases of myocardial infarction co-occur with a passage of weather fronts, compared to only 6% observed during periods of calm weather linked with high pressure [39]. A survey conducted by Baranowska showed that in the respondents’ opinion, meteoropaths are ad-versely affected by weather transitions from highs to lows [6]. In a study conducted by Höppe, individuals sensitive to weather, living in northern Germany, negatively assessed stormy weather (30%) and influx of cold air (29%). On the other hand, inhabitants of south-ern Germany experience mood decline accompanying influx of warm (30%) rather than cold air (18%). This difference may be associated with the quite frequent foehn winds in southern Germany [40]. Mackensen et al. investigated effects of weather in the sense of well-being observed in people living in Germany and in Canada. The authors reported weather sensitivity in 54.5% of the subjects representing German population while in Canada the rate amounted to 61% [12]. Likewise, Shiue et al. found a relationship between weather conditions, including wind speed, and hospital admissions due to heart disease in Germany [41]. The same relationship was shown by studies conducted in Switzerland [42] and Lithuania [16].

It is difficult to clearly establish associations between meteorological or synoptic factors and decline in people’s mood or well-being. The reasons for that may lie e.g. in varied responses to weather related stimuli, or in synergistic effects produced by various meteorological factors. Therefore, results obtained by various researchers frequently cannot be compared due to the fact that different methodologies are applied in studies.

Reviewer 2 Report

The study by Maciejczak and colleagues examined the effect of the Polish version of a Foehn wind, the “halny”, on the incidence of myocardial infarction. The concept is interesting; however, the execution does not take into account several important points.

(1) The analysis is perhaps a little naïve. Such analyses are typically performed using a time-series approach; for example, Bhaskaran et al. Int J Epidemiol 2013;42:1187 and Panagistakos et al. Int J Cardiol 2004;94:229. “Lags” are easily incorporated and so questions such as does the environmental changes coincide or predate the event examined can be determined. It would be useful to show the weekly distribution of the events. I recommend that the authors seek the assistance of a biostatistician.

(2) I am concerned too that the study is underpowered to detect what are likely to be fairly subtle differences. The authors did not appear to have run a power analysis prior to performing the study. Again, a biostatistician’s help would be useful.

(3) One element that should be examined is the question of “harvesting”; for example, Schwartz Am J Epidemiol 2000;151:440. That is, does the environmental effect simply bring forward an event that would have happened anyway – was the incidence of myocardial infarction lower than expected several days after the halny?

(4) Similarly, the influence of other weather conditions and external factors suggested to be involved should be adjusted for; for example, temperature and day of the week.

(5) Do the authors have any information on the onset-of-symptoms? Obviously, if symptoms began before the day of admission, it could affect the results.

(6) Did the authors rely solely on administrative data or did they examine the patients’ charts too; that is, was a “gold standard” used – troponins, EKG changes? Was there any specific definition of MI? Did you include STEMI and NSTEMI, was there a difference in occurrence?

(7) The authors should be careful with their labels of Foehn winds. For example, they imply that Santa Ana winds in Southern California are Foehn, when they are actually Katabatic winds. Similarly, Sirocco and Chamsin are different types of wind and these have nothing to do with the leeward side of mountain ranges. Although it appears that all these different winds have been implicated with physiological effects, they do have different origins and potentially then different consequences. It is not helpful, and is potentially misleading, to list (page 2) lots of different wind names and imply they are all the same. I think what is missing is a precise definition of “halny”; for example, pressure and temperature changes etc. Along similar lines, I imagine there are days when there might be a “weak halny” – again, a precise definition would help considerably.

(8) It appears odd to introduce a concept such as “meteoropaths” for the first time in the Discussion. The statement that they represent 50-70% of the population is very provocative. This should be supported by references.

Author Response

Response to Reviewer 2 Comments

Dear Reviewer,        

We thank you for reviewing our article titled, “Impact of foehn wind and related environmental variables on the incidence of cardiac events”. We have made every effort to improve our manuscript, as guided by the reviewer’s helpful suggestions.

We thank the reviewer of all the comments. Answers are summarised below. All changes are highlighted as red text in the manuscript.

We hope you will be pleased with the changes, and support the publication of our revised manuscript.

With kind regards,

The authors of the article

Point 1: The study by Maciejczak and colleagues examined the effect of the Polish version of a Foehn wind, the “halny”, on the incidence of myocardial infarction. The concept is interesting; however, the execution does not take into account several important points.

(1) The analysis is perhaps a little naïve. Such analyses are typically performed using a time-series approach; for example, Bhaskaran et al. Int J Epidemiol 2013;42:1187 and Panagistakos et al. Int J Cardiol 2004;94:229. “Lags” are easily incorporated and so questions such as does the environmental changes coincide or predate the event examined can be determined. It would be useful to show the weekly distribution of the events. I recommend that the authors seek the assistance of a biostatistician.

Response 1: Thank you for this valuable comment. In accordance with the Reviewer’s suggestion, we have conducted additional analysis in which we used a time-series approach. We have formulated an additional research question: do the environmental changes coincide with or predate the event examined; and we have added a two-week distribution of the events.

BEFORE

lines 2-3: Impact of foehn wind on the incidence of cardiac events

lines 16-30: Abstract: In Poland there is no data related to the impact of halny wind on the incidence of cardiac events. We decided to investigate the relationship between this weather phenomenon and the incidence of cardiac events in the population of southern Poland, a region affected by this type of wind. We examined data related to 465 patients admitted to the cardiology ward in a large regional hospital during 12 months of 2011 due to acute myocardial infarction. All the patients in the study group lived in areas affected by halny wind and at the time of the event were staying there. The frequency of admissions on halny days did not differ significantly from the admissions on the remaining days of the year (p= 0.496). No statistically significant differences were found between the number of admissions on halny days and on the remaining days during halny months (p= 0.084). We have identified a difference in the number of admissions between days with no halny and days immediately preceding onset of halny (p=0.001). During 24 hours preceding onset of halny wind, incidence of acute myocardial infarction was significantly higher than during the remaining days of the year. On the days with halny wind, incidence of cardiac events is similar to that on the remaining days of the year.

Keywords: myocardial infarction; foehn wind; halny

lines 70-71: The aim of this study was to examine the impact of foehn wind on the incidence of cardiac events in the population inhabiting a region affected by halny wind.

lines 89-94: 2.3. Data analyses

We examined the number of cardiac incidents occurring (i) on the days with halny wind blowing, and (ii) on days immediately preceding those with halny. We compared these numbers to those occurring on the remaining days of the year. The statistical analyses were performed using Calc programme, by applying chi-square test of independence and test for two indicators of structure. Statistical significance was assumed at p<0.05

lines 152-156: Based on a review of the literature it can be concluded that in many geographical regions of the world a relationship was found between weather conditions, including foehn winds, and frequency of hospital admission due to myocardial infarction [15-20]. However, so far in Poland there is no data related to the impact of foehn wind on the incidence of cardiac events.  This observation provided motivation for the current study.

lines 224-228: 6. Conclusions

  1. During 24 hours preceding onset of halny wind, incidence of acute myocardial infarction was significantly higher than during the remaining days of the year.
  2. On the days with halny wind, incidence of cardiac events is similar to that on the remaining days of the year.

AFTER

lines 2-3: Impact of foehn wind and related environmental variables on the incidence of cardiac events

lines 16-31: Abstract: In Poland there is no data related to the impact of halny wind and the related environmental variables on the incidence of cardiac events. We decided to investigate the relationship between this weather phenomenon as well as the related environmental variables and the incidence of cardiac events in the population of southern Poland, a region affected by this type of wind. We also decided to determine whether the environmental changes coincide with or predate the event examined. We analysed data related to 465 patients admitted to the cardiology ward in a large regional hospital during 12 months of 2011 due to acute myocardial infarction. All the patients in the study group lived in areas affected by halny wind and at the time of the event were staying there. The frequency of admissions on halny days did not differ significantly from the admissions on the remaining days of the year (p= 0.496). No statistically significant differences were found between the number of admissions on halny days and on the remaining days during halny months (p= 0.084). We have identified a difference in the number of admissions between days with no halny and days immediately preceding onset of halny (p=0.001), however no effects of the related environmental variables have been observed in the incidence of cardiac events (p=0.866, F=0.37). On the days with halny wind, incidence of cardiac events is similar to that on the remaining days of the year.

Keywords: myocardial infarction; foehn wind; halny; environmental variables

lines 93-96: The aim of this study was to examine the impact of the foehn wind and the related environmental variables on the incidence of cardiac events in the population inhabiting a region affected by halny wind. We also decided to determine whether the environmental changes coincide with or predate the event examined?

lines 124-132: 2.3. Data analyses

We examined the number of cardiac incidents occurring (i) on the days with halny wind blowing, and (ii) on days immediately preceding onset of halny. We compared these numbers to those occurring on the remaining days of the year. The statistical analyses were performed using Calc programme, by applying chi-square test of independence and test for two indicators of structure. Relationship between environmental variables and number of cardiac events was examined using multiple regression analysis. Distributions of mean daily air temperature, atmospheric humidity, speed of wind, pressure and total daily precipitation were shown in linear graphs. Statistical significance was assumed at p<0.05.

lines 186-235: 3.4. Assessment of the association between environmental variables and number of cardiac events

The analyses did not confirm the regression model which assumed that the different environmental variables would impact the number of all cardiac events observed on all the days of the halny months (p=0.866, F=0.37). This regression model was only explained at a level of 11.0%.

Table 6. Assessment of the number of cardiac events/ number of admissions relative to the environmental variables on all days of the halny months.

Environmental variables vs number of cardiac events

Multiple regression

R2

Adjusted R2

F

p for regression model

b

Partial correlation

p

Mean air temperature (°C)

0.11

0.01

0.37

0.866

0.01

0.04

0.586

Mean atmospheric humidity (%)

-0.00

-0.01

0.873

Mean speed of wind (m/s)

-0.00

-0.04

0.613

Pressure (hPa)

-0.01

-0.07

0.413

Total daily precipitation (l/m3)

0.00

0.00

0,962

R2- regression model. Adjusted R2 – regression model disregards extreme values. F – result of Fisher’s test. b – regression coefficient. Partial correlation – the variable Xi is correlated with the variable Y after effect of all the other independent variables is taken into account (concurrent effect of all the four factors). p– significance level

Likewise, the analyses did not confirm the regression model for a shorter period of time (multiple regression analysis for 14 days of December), where it was assumed that different environmental variables would be related to the number of cardiac events observed (p=0.451, F=1.05). This regression model was explained at a level of 39.0%, and adjusted at a level of 2.0%.

Table 7. Assessment of the number of cardiac events/ number of admissions relative to the environmental variables from 1 to 14 December.

Environmental variables vs number of cardiac events

Multiple regression

R2

Adjusted R2

F

p for regression model

b

partial correlation

p

Mean air temperature (°C)

0.39

0.02

1.05

0.451

0.06

0.11

0.753

Mean atmospheric humidity (%)

0.05

0.38

0.283

Mean speed of wind (m/s)

0.68

0.45

0.197

Pressure (hPa)

0.00

0.01

0,970

Total daily precipitation (l/m3)

-0.11

-0.22

0.546

R2- regression model. Adjusted R2 – regression model disregards extreme values. F – result of Fisher’s test. b – regression coefficient. Partial correlation – the variable Xi is correlated with the variable Y after effect of all the other independent variables is taken into account (concurrent effect of all the four factors). p– significance level

3.5. Time series data showing daily levels of environmental variables and daily number of cardiac events in the period from 1 to 14 December 2011.

Table 8 presents distributions of mean daily air temperature, atmospheric humidity, speed of wind, pressure and total daily precipitation and daily number of cardiac events, in the period from 1 to 14 December.

Table 8. Example rows of time series data from the Tarnów dataset showing daily levels of environmental variables and daily number of cardiac events.

Date

Mean air temperature
(°C)

Mean atmospheric humidity
(%)

Mean speed of wind
(m/s)

Pressure

(hPa)

Total daily precipitation
(l/m3)

N cardiac events

1 Dec. 11

4.20

85.00

1.30

1026.80

0.00

2

2 Dec.11

4.00

73,90

2,90

1016.80

0.00

2

3 Dec.11

3.10

83.00

1.40

1014,90

0.20

1

4 Dec.11

8.10

71.10

2.60

1005.10

0.20

3

5 Dec.11

7.10

76.60

2.50

999.10

1,90

2

6 Dec.11

2.00

72.50

0,90

1009.10

0.00

0

7 Dec.11

2.40

80.30

2.50

1008.30

3,90

3

8 Dec.11

2,90

87.40

2,90

1011.00

4.40

2

9 Dec.11

2,90

71,90

2,90

1014.20

6.50

1

10 Dec.11

2.40

88.10

0,90

1013.80

0.30

1

11 Dec.11

0.20

80.30

0.80

1021.00

0.00

2

12 Dec.11

3.10

81,90

1.50

1016.10

1,90

1

13 Dec.11

4.70

92.60

1.10

1014.30

0.40

2

14 Dec.11

6.80

73.60

3.00

1012.40

0.00

2

Distributions of mean daily air temperature, atmospheric humidity, speed of wind, pressure and total daily precipitation and number of cardiac events were shown using linear graphs for the period from 1 to 14 December 2011 (Figure 1 and 2).

Figure 1. Distributions of mean daily air temperature, atmospheric humidity and speed of wind for the period from 1 to 14 December 2011.

Figure 2. Distributions of mean pressure, total daily precipitation and number of cardiac events for the period from 1 to 14 December 2011.

Analysis of the environmental variables (i.e. air temperature, atmospheric humidity, speed of wind, pressure and total daily precipitation) failed to demonstrate that any of these variables, or all of them jointly, affected the incidence of cardiac events on the days with halny or on the days preceding its onset.

lines 236-275: 4. Discussion

Based on a review of the literature it can be concluded that many researchers are investigating the association between wind-related environmental variables, and cardio- or cerebrovascular events, including the impact observed on days preceding onset of such weather conditions [14-23]. However, so far in Poland no such studies have been published yet. Boussoussou et al. have reported positive significant association between acute cardiovascular diseases and a cold front effect lagged by one day [14]. Mohammad et al. demonstrated that incidence of myocardial infarction increased with higher wind velocity, lower air temperature, lower atmospheric air pressure, and shorter sunshine duration [15]. Similarly, Honda et al. found that increased incidence of acute myocardial infarction corresponds to lower minimum temperature occurring on the 2nd day preceding the onset of wind [17]. Radišauskas et al. establish that myocardial infarction morbidity is directly correlated to wind speed, and atmospheric pressure and inversely correlated to atmospheric temperatures, and rainfall level [16]. Goerre et al. reported that the incidence of myocardial infarctions is significantly affected by wind activity, ambient pressure, and pressure gradient [15]. Vencloviene et al. showed that prognostic factors for the risk of acute coronary syndromes include such environmental variables as wind speed, daily air temperature, and barometric pressure on the day of the onset and on the two days preceding the ambulance call [18]. Likewise, Ezekowitz et al. showed relationships between the likelihood of an acute coronary event and rapid changes in weather conditions (including excessively frequent increases in wind speed) [19]. Hensel et al. showed that increasing wind speed corresponds to greater likelihood of out-of-hospital cardiac arrest of presumed cardiac aetiology [20]. Vencloviene et al. demonstrated that increased risk of paroxysmal atrial fibrillation occurring at night was linked with wind speed above the median (lag 2-4 days) [21]. Tamasauskiene et al. pointed out that a greater risk of ischemic stroke in older patients was associated with high wind speed on the day of the incident [22]. Similarly, Kim et al. showed that the odds ratios for wind speed, daily wind speed range, and wind chill for patients with ischemic stroke were slightly greater than in the total sample [23].

Effects of seasonal changes in environmental variables have also been pointed out. According to Gallerani et al., it is possible to observe 6-month-period regularities in the incidence of subarachnoid haemorrhage, spring and autumn being the peak seasons. The authors propose an explanation that during these transitional seasons the arterial wall is exposed to frequent stress episodes due to typical changes in temperature and that may result in aneurysmal rupture occurring more frequently than e.g. during the more stable periods of winter cold [26]. Investigators from the International Worldwide Registry reported that the high incidence of acute aortic dissection in winter is observed in various climate settings (both cold and temperate), which seems to suggest that the major mechanistic factors contributing to the phenomenon include the relative change in temperature, rather than absolute temperature, and/or circannual rhythms [27].

Contrary to the findings reported by other researchers [14-23], the current study did not confirm effects of environmental factors investigated, i.e. air temperature, atmospheric humidity, speed of wind, pressure and total daily precipitation in the incidence of cardiac events, either on halny days, or on days preceding the onset of halny.

Lines 346-357: 6. Conclusions

  1. On the days with halny wind, incidence of cardiac events is similar to that on the remaining days of the year.
  2. The findings show no effect of the related environmental variables on the number of cardiac events observed.
  3. During 24 hours preceding onset of halny wind, incidence of acute myocardial infarction was significantly higher than during the remaining days of the year, however no evidence was found to confirm impact of the related environmental variables on the number of admissions during the relevant period.
  4. To establish a cause-and-effect relationship, more comprehensive data should be acquired from a larger number of regions affected by halny wind in Poland, and the relevant observations should be continued for a period longer than one calendar year.

(2) I am concerned too that the study is underpowered to detect what are likely to be fairly subtle differences. The authors did not appear to have run a power analysis prior to performing the study. Again, a biostatistician’s help would be useful.

Response 2: Thank you for the helpful suggestions. We have added sample size calculation.

BEFORE

-

AFTER

lines 134-140: 2.4. Sample Size

The minimum size of the sample was determined using a sample size calculator, taking into account the number of individuals admitted to the cardiology ward due to acute myocardial infarction, i.e. approximately 465 patients. A fraction size of 0.8 was used, with a maximum error of 5%, a sample size of 161 patients was obtained. In our study we analysed data related to 227 individuals admitted on halny days, one day before onset of halny and during the remaining days of the relevant months.

(3) One element that should be examined is the question of “harvesting”; for example, Schwartz Am J Epidemiol 2000;151:440. That is, does the environmental effect simply bring forward an event that would have happened anyway – was the incidence of myocardial infarction lower than expected several days after the halny?

Response 3: Thank you for this valuable comment. Analysis of the environmental variables (i.e. air temperature, atmospheric humidity, speed of wind, pressure and total daily precipitation) failed to demonstrate that any of these variables, or all of them jointly, affected the incidence of cardiac events on the days with halny or on the days preceding its onset.

BEFORE

-

AFTER

lines 186-235: 3.4. Assessment of the association between environmental variables and number of cardiac events

The analyses did not confirm the regression model which assumed that the different environmental variables would impact the number of all cardiac events observed on all the days of the halny months (p=0.866, F=0.37). This regression model was only explained at a level of 11.0%.

Table 6. Assessment of the number of cardiac events/ number of admissions relative to the environmental variables on all days of the halny months.

Environmental variables vs number of cardiac events

Multiple regression

R2

Adjusted R2

F

p for regression model

b

Partial correlation

p

Mean air temperature (°C)

0.11

0.01

0.37

0.866

0.01

0.04

0.586

Mean atmospheric humidity (%)

-0.00

-0.01

0.873

Mean speed of wind (m/s)

-0.00

-0.04

0.613

Pressure (hPa)

-0.01

-0.07

0.413

Total daily precipitation (l/m3)

0.00

0.00

0,962

R2- regression model. Adjusted R2 – regression model disregards extreme values. F – result of Fisher’s test. b – regression coefficient. Partial correlation – the variable Xi is correlated with the variable Y after effect of all the other independent variables is taken into account (concurrent effect of all the four factors). p– significance level

Likewise, the analyses did not confirm the regression model for a shorter period of time (multiple regression analysis for 14 days of December), where it was assumed that different environmental variables would be related to the number of cardiac events observed (p=0.451, F=1.05). This regression model was explained at a level of 39.0%, and adjusted at a level of 2.0%.

Table 7. Assessment of the number of cardiac events/ number of admissions relative to the environmental variables from 1 to 14 December.

Environmental variables vs number of cardiac events

Multiple regression

R2

Adjusted R2

F

p for regression model

b

partial correlation

p

Mean air temperature (°C)

0.39

0.02

1.05

0.451

0.06

0.11

0.753

Mean atmospheric humidity (%)

0.05

0.38

0.283

Mean speed of wind (m/s)

0.68

0.45

0.197

Pressure (hPa)

0.00

0.01

0,970

Total daily precipitation (l/m3)

-0.11

-0.22

0.546

R2- regression model. Adjusted R2 – regression model disregards extreme values. F – result of Fisher’s test. b – regression coefficient. Partial correlation – the variable Xi is correlated with the variable Y after effect of all the other independent variables is taken into account (concurrent effect of all the four factors). p– significance level

3.5. Time series data showing daily levels of environmental variables and daily number of cardiac events in the period from 1 to 14 December 2011.

Table 8 presents distributions of mean daily air temperature, atmospheric humidity, speed of wind, pressure and total daily precipitation and daily number of cardiac events, in the period from 1 to 14 December.

Table 8. Example rows of time series data from the Tarnów dataset showing daily levels of environmental variables and daily number of cardiac events.

Date

Mean air temperature
(°C)

Mean atmospheric humidity
(%)

Mean speed of wind
(m/s)

Pressure

(hPa)

Total daily precipitation
(l/m3)

N cardiac events

1 Dec. 11

4.20

85.00

1.30

1026.80

0.00

2

2 Dec.11

4.00

73,90

2,90

1016.80

0.00

2

3 Dec.11

3.10

83.00

1.40

1014,90

0.20

1

4 Dec.11

8.10

71.10

2.60

1005.10

0.20

3

5 Dec.11

7.10

76.60

2.50

999.10

1,90

2

6 Dec.11

2.00

72.50

0,90

1009.10

0.00

0

7 Dec.11

2.40

80.30

2.50

1008.30

3,90

3

8 Dec.11

2,90

87.40

2,90

1011.00

4.40

2

9 Dec.11

2,90

71,90

2,90

1014.20

6.50

1

10 Dec.11

2.40

88.10

0,90

1013.80

0.30

1

11 Dec.11

0.20

80.30

0.80

1021.00

0.00

2

12 Dec.11

3.10

81,90

1.50

1016.10

1,90

1

13 Dec.11

4.70

92.60

1.10

1014.30

0.40

2

14 Dec.11

6.80

73.60

3.00

1012.40

0.00

2

Distributions of mean daily air temperature, atmospheric humidity, speed of wind, pressure and total daily precipitation and number of cardiac events were shown using linear graphs for the period from 1 to 14 December 2011 (Figure 1 and 2).

Figure 1. Distributions of mean daily air temperature, atmospheric humidity and speed of wind for the period from 1 to 14 December 2011.

Figure 2. Distributions of mean pressure, total daily precipitation and number of cardiac events for the period from 1 to 14 December 2011.

Analysis of the environmental variables (i.e. air temperature, atmospheric humidity, speed of wind, pressure and total daily precipitation) failed to demonstrate that any of these variables, or all of them jointly, affected the incidence of cardiac events on the days with halny or on the days preceding its onset.

(4) Similarly, the influence of other weather conditions and external factors suggested to be involved should be adjusted for; for example, temperature and day of the week.

Response 4: Thank you for the helpful suggestions. In accordance with the Reviewer’s suggestion we have adjusted the influence of other environmental factors involved (as above).

(5) Do the authors have any information on the onset-of-symptoms? Obviously, if symptoms began before the day of admission, it could affect the results.

Response 5: Thank you for this valuable comment. Patients taken into account in the study presented evident symptoms of heart disease which started within 12 hours before admission. Patients with similar symptoms which began earlier than 12 hours prior to admission were not included in the study group.

BEFORE

lines 74-78: We examined a consecutive series of patients with acute myocardial infarction classified according to ICD-10 in the medical records as I21 (the whole subgroup of I21.0 – I21.4 and I21.9), admitted to the cardiology ward in a large regional hospital during 12 months of 2011 [9]. All the patients in the study group lived in areas affected by halny wind and at the time of the event were staying there. The hospital is also located in an area affected by halny.

AFTER

lines 101-113: We examined a consecutive series of patients with acute myocardial infarction classified according to ICD-10 in the medical records as I21 (the whole subgroup of I21.0 – I21.4 and I21.9), admitted to the cardiology ward in a large regional hospital during 12 months of 2011 [24]. Patients taken into account in the study presented evident symptoms of heart disease which started within 12 hours before admission. Patients with similar symptoms which began earlier than 12 hours prior to admission were not included in the study group. The criteria used in the diagnosis of acute myocardial infarction included: 1) STEMI; 2) NSTEMI with elevated troponin levels, exceeding the norm adopted by our hospital; the elevated concentrations were confirmed in two consecutive tests carried out 3-4 and 9-12 hours after the first measurement. The diagnosis of acute myocardial infarction in majority of the patients (65%) was based on the first criterion. Troponin levels were also measured in the patients with STEMI, as a way to monitor progress of the infarction. All the patients in the study group lived in areas affected by halny wind and at the time of the event were staying there. The hospital is also located in an area affected by halny.

(6) Did the authors rely solely on administrative data or did they examine the patients’ charts too; that is, was a “gold standard” used – troponins, EKG changes? Was there any specific definition of MI? Did you include STEMI and NSTEMI, was there a difference in occurrence?

Response 6: Thank you for these helpful suggestions. Yes, the patients’ medical records were examined. The criteria used in the diagnosis of acute myocardial infarction included:

  1. STEMI
  2. NSTEMI with elevated troponin levels, exceeding the norm adopted by our hospital; the elevated concentrations were confirmed in two consecutive tests carried out 3-4 and 9-12 hours after the first measurement.

The diagnosis of acute myocardial infarction in majority of the patients (65%) was based on the first criterion. Troponin levels were also measured in the patients with STEMI, as a way to monitor progress of the infarction.

BEFORE

lines 74-78: We examined a consecutive series of patients with acute myocardial infarction classified according to ICD-10 in the medical records as I21 (the whole subgroup of I21.0 – I21.4 and I21.9), admitted to the cardiology ward in a large regional hospital during 12 months of 2011 [9]. All the patients in the study group lived in areas affected by halny wind and at the time of the event were staying there. The hospital is also located in an area affected by halny.

AFTER

lines 101-113: We examined a consecutive series of patients with acute myocardial infarction classified according to ICD-10 in the medical records as I21 (the whole subgroup of I21.0 – I21.4 and I21.9), admitted to the cardiology ward in a large regional hospital during 12 months of 2011 [24]. Patients taken into account in the study presented evident symptoms of heart disease which started within 12 hours before admission. Patients with similar symptoms which began earlier than 12 hours prior to admission were not included in the study group. The criteria used in the diagnosis of acute myocardial infarction included: 1) STEMI; 2) NSTEMI with elevated troponin levels, exceeding the norm adopted by our hospital; the elevated concentrations were confirmed in two consecutive tests carried out 3-4 and 9-12 hours after the first measurement. The diagnosis of acute myocardial infarction in majority of the patients (65%) was based on the first criterion. Troponin levels were also measured in the patients with STEMI, as a way to monitor progress of the infarction. All the patients in the study group lived in areas affected by halny wind and at the time of the event were staying there. The hospital is also located in an area affected by halny.

(7) The authors should be careful with their labels of Foehn winds. For example, they imply that Santa Ana winds in Southern California are Foehn, when they are actually Katabatic winds. Similarly, Sirocco and Chamsin are different types of wind and these have nothing to do with the leeward side of mountain ranges. Although it appears that all these different winds have been implicated with physiological effects, they do have different origins and potentially then different consequences. It is not helpful, and is potentially misleading, to list (page 2) lots of different wind names and imply they are all the same. I think what is missing is a precise definition of “halny”; for example, pressure and temperature changes etc. Along similar lines, I imagine there are days when there might be a “weak halny” – again, a precise definition would help considerably.

Response 7: Thank you for the helpful comment. In accordance with the Reviewer’s suggestion we have removed misleading information and added a precise definition of “halny”.

BEFORE

lines 47-69: A local variety of foehn, “halny” wind is produced by violent flow of air masses from the southern side of the Carpathians or the Sudetes. This happens when Poland is approached from the west by a deep cyclone (low-pressure area), while the centre of anticyclone (high-pressure) is in Ukraine [6,7]. In addition to an abrupt change in temperature and large speed of the wind, there are also other consequences of halny wind. The first one involves rapid pressure drop, and the other one is linked with increased condensation of positive ions in the air as a result of which microscopic particles of gases and aerosols, including harmful elements and chemical compounds, can easily penetrate organisms. This may lead to headaches, muscle and join pain, increased problems related to the cardiovascular and nervous systems, and most commonly to poor mood [6-8].

In the literature we can find reports discussing health-related changes in response to foehn wind. It is believed that foehn wind may also lead to increased aggressiveness, more severe depressive states, emotional crises ending with suicide attempts, agitation, anxiety, fatigue, weariness, problems with concentration, sleep disorders, migraine, muscle and joint pain as well as cardiovascular incidents [3, 4, 6, 8].  The foehn wind known as “halny”, occurring in southern Poland, has equally bad reputation as the foehn winds blowing in other parts of Europe and the world. We decided to investigate its impact on the incidence of cardiac events in the population of southern Poland, a region affected by this type of wind. The present study was encouraged by findings reported in biometeorological literature showing negative effects of other foehn winds, such as "Chinook" descending the slopes of the Rocky Mountains, "Santa Ana" and "Santa Maria" blowing in California, "Megas" in Greece, and "Zonda" blowing in the Andes. Negative effect of this type of wind is sometimes considered as a mitigating circumstance in judicial decisions related to minor offences, e.g. in Switzerland and in the Mediterranean countries affected by sirocco and chamsin winds [4,5].

AFTER

lines 47-66: A local variety of foehn, “halny” wind is produced by violent flow of air masses from the southern side of the Carpathians or the Sudetes. This happens when Poland is approached from the west by a deep cyclone (low-pressure area), while the centre of anticyclone (high-pressure) is in Ukraine [3,4]. In addition to an abrupt change in temperature and large speed of the wind, there are also other consequences of halny wind. The first one involves rapid pressure drop, and the other one is linked with increased condensation of positive ions in the air as a result of which microscopic particles of gases and aerosols, including harmful elements and chemical compounds, can easily penetrate organisms [3,4]. Halny wind frequently leads to considerable increase in the temperature and decrease in humidity on the leeward side of the mountains. The strong abrupt gusts of wind at times reach a speed typical for hurricane, and are accompanied with characteristic calm intervals. The foehn winds are observed in the Tatras rather frequently, particularly during the cold seasons. The criteria commonly applied by Polish researchers investigating foehn winds occurring in the mountainous areas include: direction (usually 90-270°) and wind speed (> 4 m/s, commonly ≥10 m/s, at times increasing to even 60 m/s), occurrence of strong gusts and calm intervals, sudden rise in the air temperature (by several degrees in a short time) and decrease in the humidity (commonly ≤70%), changes in atmospheric pressure and / or certain types of the clouds. In foothill and lowland areas, the wind is dry and warm [4].

It is believed that foehn wind may lead to increased aggressiveness, more severe depressive states, emotional crises ending with suicide attempts, agitation, anxiety, fatigue, weariness, muscle and joint pain, problems with concentration, sleep disorders, migraine, as well as headaches [4-7]. 

(8) It appears odd to introduce a concept such as “meteoropaths” for the first time in the Discussion. The statement that they represent 50-70% of the population is very provocative. This should be supported by references.

Response 8: Thank you for the helpful suggestions. We have moved the relevant information from the discussion to introduction and we have used supporting references.

BEFORE

lines 140-151: 4. Discussion

People with high sensitivity to weather changes are referred to as meteoropaths. They present increased tension of the vegetative system and poor adaptative responses. Effects of weather fronts, in particular cold fronts, include decrease in the temperature and increase in the atmospheric pressure, responses to which include vasoconstriction, potentially posing a burden for the circulatory system [11-13] Nowadays, it is estimated that the number of me-teoropaths is consistently growing. They constitute approximately 50%-70% of the healthy populations. It is also believed that meteoropathy is a “disease” of affluence, caused by stress, work overload, and unhealthy lifestyles, since meteoropaths mainly include residents of urban areas, whose immunity is affected by the long time spent in the artificial microcli-mates of their homes and offices, by a lack of exercise and by polluted air. It is also pointed out that meteoropathy is most commonly observed in individuals affected by coronary dis-ease, hypertension and ulcer disease [11-14].

AFTER

lines 67-80: 1. Introduction

People with high sensitivity to weather changes are referred to as meteoropaths. They present increased tension of the vegetative system and poor adaptative responses. Effects of weather fronts, in particular cold fronts, include decrease in the temperature and increase in the atmospheric pressure, responses to which include vasoconstriction, potentially posing a burden for the circulatory system [8-10] Nowadays, it is estimated that the number of meteoropaths is consistently growing. According to the data from the Institute of Meteorology and Water Management, as many as 50-70% of Poles respond to weather changes [11]. Similar statistics have been published in other countries, e.g. in Germany 54.5% of the population is weather sensitive, in Canada the relevant rate amounts to 61% [12]. It is also believed that meteoropathy is a “disease” of affluence, caused by stress, work overload, and unhealthy lifestyles, since meteoropaths mainly include residents of urban areas, whose immunity is affected by the long time spent in the artificial microclimates of their homes and offices, by a lack of exercise and by polluted air. It has also been pointed out that meteoropathy is most commonly observed in individuals affected by coronary disease, hypertension and ulcer disease [8-10,13].

Reviewer 3 Report

Major remarks:

p1, line 36-37: There relevance of references 3-8 are questionable. Reference 3 discusses the causes of foehn warming, and nothing about being a potential risk factor for cardiac incidents.  Reference 5 mentions the vague concept “foehn sickness”, how this wind can affect the human energy budget, and how it may make people feel unpleasant ⸺absolutely nothing about incidence of cardiac events.  Reference 6 discusses the damaging effect of increased wind speed and temperature on forests and infrastructure, not human medical conditions. Similarly, reference 7 makes no connection between foehn wind and human medical conditions.  Only reference 8 assess the human impact, but this is limited to migraines, not cardiac events.  References 15-20 (only mentioned in the discussion) may be relevant, but I am not convinced that these would be sufficient to make a case for investigating the effects of foehn winds specifically as opposed to weather more generally.

The problems undermining the merit of this paper is actually articulated by the authors on p5, lines 152, 156.  First, the literature (which should be in the introduction, not the discussion) which supports the notion there may be a connection between cardiac events and foehn winds is entirely lacking.  The supposed conclusion “in many geographical regions of the world a relationship was found between weather conditions… and myocardial infarction” is unsupported. Thus, it is not surprising that there is no data relating to foehn wind and cardiac events in Poland.

In addition to the lack of evidence and motivation from the literature, the analysis is completely flawed.  The analysis focuses solely on correlation or differences between groups (which appears to be lacking) which is then used to make statements about causation.  Even if hospital admissions were somewhat higher on halny days, it seems logical that this would be a causal effect – the lack of literature supporting this view speaks volumes.  The idea of then trying to determine the causal effect of halny on cardiac related hospital admissions on days BEFORE a halny is absurd.  If the authors wanted to do a serious analysis, rather than simply grouping all admissions on a particular type of day (halny, pre-halny, etc.) and comparing difference between groups, the authors should consider a longitudinal study and look at the temporal autocorrelation between admissions and weather events.  But first, the literature should be revisited to make a case for why a causal relationship between foehn/halny wind (not weather more generally) and cardiac events (not other health conditions like headaches) is even rational.

Minor remarks:

p3, line 99-101: The number of male patients is observed, so do the authors mean that this was “confirmed in the test for significance”?

p5 line 141: This idea that some people are sensitive to weather changes (meteoropaths) seems important and should be cited in the introduction, not in the discussion.  The lack of supporting literature completely undermines the objective of this manuscript.

Author Response

Response to Reviewer 3 Comments

Dear Reviewer,        

We thank you for reviewing our article titled, “Impact of foehn wind and related environmental variables on the incidence of cardiac events”. We have made every effort to improve our manuscript, as guided by the reviewer’s helpful suggestions.

We thank the reviewer of all the comments. Answers are summarised below. All changes are highlighted as red text in the manuscript.

We hope you will be pleased with the changes, and support the publication of our revised manuscript.

With kind regards,

The authors of the article

Point 1: Major remarks:

p1, line 36-37: There relevance of references 3-8 are questionable. Reference 3 discusses the causes of foehn warming, and nothing about being a potential risk factor for cardiac incidents.  Reference 5 mentions the vague concept “foehn sickness”, how this wind can affect the human energy budget, and how it may make people feel unpleasant ⸺absolutely nothing about incidence of cardiac events.  Reference 6 discusses the damaging effect of increased wind speed and temperature on forests and infrastructure, not human medical conditions. Similarly, reference 7 makes no connection between foehn wind and human medical conditions.  Only reference 8 assess the human impact, but this is limited to migraines, not cardiac events.  References 15-20 (only mentioned in the discussion) may be relevant, but I am not convinced that these would be sufficient to make a case for investigating the effects of foehn winds specifically as opposed to weather more generally.

The problems undermining the merit of this paper is actually articulated by the authors on p5, lines 152, 156.  First, the literature (which should be in the introduction, not the discussion) which supports the notion there may be a connection between cardiac events and foehn winds is entirely lacking.  The supposed conclusion “in many geographical regions of the world a relationship was found between weather conditions… and myocardial infarction” is unsupported. Thus, it is not surprising that there is no data relating to foehn wind and cardiac events in Poland.

In addition to the lack of evidence and motivation from the literature, the analysis is completely flawed.  The analysis focuses solely on correlation or differences between groups (which appears to be lacking) which is then used to make statements about causation.  Even if hospital admissions were somewhat higher on halny days, it seems logical that this would be a causal effect – the lack of literature supporting this view speaks volumes.  The idea of then trying to determine the causal effect of halny on cardiac related hospital admissions on days BEFORE a halny is absurd.  If the authors wanted to do a serious analysis, rather than simply grouping all admissions on a particular type of day (halny, pre-halny, etc.) and comparing difference between groups, the authors should consider a longitudinal study and look at the temporal autocorrelation between admissions and weather events.  But first, the literature should be revisited to make a case for why a causal relationship between foehn/halny wind (not weather more generally) and cardiac events (not other health conditions like headaches) is even rational.

 Response 1: Thank you for this valuable comment. In accordance with the Reviewer’s suggestion, we have revisited the literature and rewritten the introduction. Additionally, we have supplemented our results to show analysis of the correlations between the number of admissions and weather events (environmental variables).

BEFORE

lines 2-3: Impact of foehn wind on the incidence of cardiac events

lines 32-71: 1. Introduction

Cardiovascular disorders are the most common cause of mortality worldwide, including most European countries, and Poland [1]. Seasonal peaks in acute myocardial infarction incidence have been widely reported. Weather has been postulated to be one of the elements greatly contributing to this association [2]. Foehn wind may be one of the weather-related factors which may provoke cardiac incidents [3-8].

The term “föhn” is used with reference to a type of strong and turbulent wind blowing down from mountain crests towards the lee side, and bringing dry and warm air to the surface. Foehn events rather frequently occur in the Alps, or more specifically on the northern side of the mountains, and constitute a significant challenge in weather forecasting for the Alpine valleys, inhabited by a lot of people. The northern Alpine foehn is frequently observed in the prefrontal environment. The foehn clearance reflects the leeward range of the foehn in the mountains, i.e. a region with no clouds where dry and warm air occurs at the surface, just as in the foehn region but with no strong winds. This meteorological phenomenon is not unique to the Alps, and it can be observed over other high mountains worldwide [7].

A local variety of foehn, “halny” wind is produced by violent flow of air masses from the southern side of the Carpathians or the Sudetes. This happens when Poland is approached from the west by a deep cyclone (low-pressure area), while the centre of anticyclone (high-pressure) is in Ukraine [6,7]. In addition to an abrupt change in temperature and large speed of the wind, there are also other consequences of halny wind. The first one involves rapid pressure drop, and the other one is linked with increased condensation of positive ions in the air as a result of which microscopic particles of gases and aerosols, including harmful elements and chemical compounds, can easily penetrate organisms. This may lead to headaches, muscle and join pain, increased problems related to the cardiovascular and nervous systems, and most commonly to poor mood [6-8].

In the literature we can find reports discussing health-related changes in response to foehn wind. It is believed that foehn wind may also lead to increased aggressiveness, more severe depressive states, emotional crises ending with suicide attempts, agitation, anxiety, fatigue, weariness, problems with concentration, sleep disorders, migraine, muscle and joint pain as well as cardiovascular incidents [3, 4, 6, 8].  The foehn wind known as “halny”, occurring in southern Poland, has equally bad reputation as the foehn winds blowing in other parts of Europe and the world. We decided to investigate its impact on the incidence of cardiac events in the population of southern Poland, a region affected by this type of wind. The present study was encouraged by findings reported in biometeorological literature showing negative effects of other foehn winds, such as "Chinook" descending the slopes of the Rocky Mountains, "Santa Ana" and "Santa Maria" blowing in California, "Megas" in Greece, and "Zonda" blowing in the Andes. Negative effect of this type of wind is sometimes considered as a mitigating circumstance in judicial decisions related to minor offences, e.g. in Switzerland and in the Mediterranean countries affected by sirocco and chamsin winds [4,5].

The aim of this study was to examine the impact of foehn wind on the incidence of cardiac events in the population inhabiting a region affected by halny wind.

lines 70-71: The aim of this study was to examine the impact of foehn wind on the incidence of cardiac events in the population inhabiting a region affected by halny wind.

lines 89-94: 2.3. Data analyses

We examined the number of cardiac incidents occurring (i) on the days with halny wind blowing, and (ii) on days immediately preceding those with halny. We compared these numbers to those occurring on the remaining days of the year. The statistical analyses were performed using Calc programme, by applying chi-square test of independence and test for two indicators of structure. Statistical significance was assumed at p<0.05

lines 140-212: 4. Discussion

People with high sensitivity to weather changes are referred to as meteoropaths. They present increased tension of the vegetative system and poor adaptative responses. Effects of weather fronts, in particular cold fronts, include decrease in the temperature and increase in the atmospheric pressure, responses to which include vasoconstriction, potentially posing a burden for the circulatory system [11-13] Nowadays, it is estimated that the number of me-teoropaths is consistently growing. They constitute approximately 50%-70% of the healthy populations. It is also believed that meteoropathy is a “disease” of affluence, caused by stress, work overload, and unhealthy lifestyles, since meteoropaths mainly include residents of urban areas, whose immunity is affected by the long time spent in the artificial microcli-mates of their homes and offices, by a lack of exercise and by polluted air. It is also pointed out that meteoropathy is most commonly observed in individuals affected by coronary dis-ease, hypertension and ulcer disease [11-14].

Based on a review of the literature it can be concluded that in many geographical regions of the world a relationship was found between weather conditions, including foehn winds, and frequency of hospital admission due to myocardial infarction [15-20]. However, so far in Poland there is no data related to the impact of foehn wind on the incidence of cardiac events.  This observation provided motivation for the current study.

Clinical studies provide evidence that some meteorological factors, and most important-ly weather changes occurring rapidly, e.g. in one day, may play an important role in induc-ing subjectively perceived indisposition in healthy individuals and in promoting disease symptoms in patients. Notably, the same weather changes may induce completely different bodily responses. As an example, a large drop in barometric pressure will induce a decrease in blood pressure in some individuals and an increase in other people [2-5, 21-24]. Due to this no uniform recommendations for dealing with sudden changes in weather can be ad-dressed e.g. to all patients with cardiovascular problems.

The current study was an attempt to collect evidence related to the impact of foehn wind on the incidence of cardiac events. The basic meteotropic factor adopted in the study was defined as halny wind and the accompanying change from cold to warm weather front. The acquired results partly provide evidence related to effects of halny wind in the incidence of acute myocardial infarction. Although the findings covering the relevant period of time do not show significantly higher number of cardiac events on halny days, such correlation was identified in the case of days immediately preceding occurrence of halny. On days preceding halny wind onset, incidence of cardiac events was higher than on the remaining days. Re-search conducted by other authors also suggests occurrence of similar synchronicities during a period ranging from 3 to 24 hours prior to an arrival of a weather front [27,28]. It has also been shown that meteorotropic responses in both healthy people and individuals affected by medical conditions may be observed more in advance of specific changes in weather; in ex-treme cases these may be “sensed” up to 48 hours before they happen. Such sensitivity may result from changes in electric field of the atmosphere, preceding arrival of a weather front [29]. It can be speculated that the greater number of cardiac incidents identified in our study 24 hours before onset of halny wind may be associated with the fact that indeed the wind is only an effect of earlier turbulent changes in atmospheric pressure systems. Hence, it is not halny wind itself that impacts cardiac events but the preceding changes in weather fronts. The factor of key importance here may be the drop in atmospheric pressure since onset of halny is determined by co-occurrence of high pressure on the southern side and low pressure on the northern side of the Tatra mountains. And this pressure drop may impact cardiac events. Other published studies confirm that changes in atmospheric pressure produce ad-verse effects in human body, possibly also leading to cardiac incidents [30-33].

As early as 1953, it was suggested by Petersen that human organism may be stimulated by specific relationships between weather fronts [4]. According to data from the Institute of Meteorology and Water Management, as many as 50-70% of Poles respond to weather changes [25]. Cardiovascular problems are most frequently listed among meteorotropic dis-eases. In Poland the earliest studies focusing on impact of weather on functioning of cardio-vascular system were carried out in the mid-1800s [4]. Research shows impact of short-term weather changes on intensity of ischemic heart disease symptoms, increased oscillations in arterial blood pressure, and higher incidence of acute myocardial infarction. Over 34% of fatal cases of myocardial infarction co-occur with a passage of weather fronts, compared to only 6% observed during periods of calm weather linked with high pressure [26]. A survey conducted by Baranowska showed that in the respondents’ opinion, meteoropaths are ad-versely affected by weather transitions from highs to lows [4]. In a study conducted by Höppe, individuals sensitive to weather, living in northern Germany, negatively assessed stormy weather (30%) and influx of cold air (29%). On the other hand, inhabitants of south-ern Germany experience mood decline accompanying influx of warm (30%) rather than cold air (18%). This difference may be associated with the quite frequent foehn winds in southern Germany [16]. Mackensen et al. investigated effects of weather in the sense of well-being observed in people living in Germany and in Canada. The authors reported weather sensitivi-ty in 54.5% of the subjects representing German population while in Canada the rate amounted to 61% [17]. Likewise, Shiue et al. found a relationship between weather condi-tions, including wind speed, and hospital admissions due to heart disease in Germany [18]. The same relationship was shown by a study conducted in Lithuania [20].

It is difficult to clearly establish associations between meteorological or synoptic factors and decline in people’s mood or well-being. The reasons for that may lie e.g. in var-ied responses to weather related stimuli, or in synergistic effects produced by various mete-orological factors. Therefore, results obtained by various researchers frequently cannot be compared due to the fact that different methodologies are applied in studies.

lines 224-228: 6. Conclusions

  1. During 24 hours preceding onset of halny wind, incidence of acute myocardial infarction was significantly higher than during the remaining days of the year.
  2. On the days with halny wind, incidence of cardiac events is similar to that on the remaining days of the year.

AFTER

lines 2-3: Impact of foehn wind and related environmental variables on the incidence of cardiac events

lines 33-96: 1. Introduction

Cardiovascular disorders are the most common cause of mortality worldwide, including most European countries, and Poland [1]. Seasonal peaks in acute myocardial infarction incidence have been widely reported. Weather has been postulated to be one of the elements greatly contributing to this association [2].

The term “föhn” is used with reference to a type of strong and turbulent wind blowing down from mountain crests towards the lee side, and bringing dry and warm air to the surface. Foehn events rather frequently occur in the Alps, or more specifically on the northern side of the mountains, and constitute a significant challenge in weather forecasting for the Alpine valleys, inhabited by a lot of people. The northern Alpine foehn is frequently observed in the prefrontal environment. The foehn clearance reflects the leeward range of the foehn in the mountains, i.e. a region with no clouds where dry and warm air occurs at the surface, just as in the foehn region but with no strong winds. This meteorological phenomenon is not unique to the Alps, and it can be observed over other high mountains worldwide [3].

A local variety of foehn, “halny” wind is produced by violent flow of air masses from the southern side of the Carpathians or the Sudetes. This happens when Poland is approached from the west by a deep cyclone (low-pressure area), while the centre of anticyclone (high-pressure) is in Ukraine [3,4]. In addition to an abrupt change in temperature and large speed of the wind, there are also other consequences of halny wind. The first one involves rapid pressure drop, and the other one is linked with increased condensation of positive ions in the air as a result of which microscopic particles of gases and aerosols, including harmful elements and chemical compounds, can easily penetrate organisms [3,4]. Halny wind frequently leads to considerable increase in the temperature and decrease in humidity on the leeward side of the mountains. The strong abrupt gusts of wind at times reach a speed typical for hurricane, and are accompanied with characteristic calm intervals. The foehn winds are observed in the Tatras rather frequently, particularly during the cold seasons. The criteria commonly applied by Polish researchers investigating foehn winds occurring in the mountainous areas include: direction (usually 90-270) and wind speed (> 4 m/s, commonly ≥10 m/s, at times increasing to even 60 m/s), occurrence of strong gusts and calm intervals, sudden rise in the air temperature (by several degrees in a short time) and decrease in the humidity (commonly ≤70%), changes in atmospheric pressure and / or certain types of the clouds. In foothill and lowland areas, the wind is dry and warm [4].

It is believed that foehn wind may lead to increased aggressiveness, more severe depressive states, emotional crises ending with suicide attempts, agitation, anxiety, fatigue, weariness, muscle and joint pain, problems with concentration, sleep disorders, migraine, as well as headaches [4-7]. 

People with high sensitivity to weather changes are referred to as meteoropaths. They present increased tension of the vegetative system and poor adaptative responses. Effects of weather fronts, in particular cold fronts, include decrease in the temperature and increase in the atmospheric pressure, responses to which include vasoconstriction, potentially posing a burden for the circulatory system [8-10] Nowadays, it is estimated that the number of meteoropaths is consistently growing. According to the data from the Institute of Meteorology and Water Management, as many as 50-70% of Poles respond to weather changes [11]. Similar statistics have been published in other countries, e.g. in Germany 54.5% of the population is weather sensitive, in Canada the relevant rate amounts to 61% [12]. It is also believed that meteoropathy is a “disease” of affluence, caused by stress, work overload, and unhealthy lifestyles, since meteoropaths mainly include residents of urban areas, whose immunity is affected by the long time spent in the artificial microclimates of their homes and offices, by a lack of exercise and by polluted air. It has also been pointed out that meteoropathy is most commonly observed in individuals affected by coronary disease, hypertension and ulcer disease [8-10,13].

A large body of evidence reported in the literature shows there is an association between wind- related environmental variables, and cardiovascular or cerebrovascular events, with impact observed also on the preceding days [14-23]. The related research findings focus on acute cardiovascular diseases and atmospheric front effect lagged by one day [14], myocardial infarction in relation to wind velocity, atmospheric air pressure, air temperature [15, 16] and lower minimum temperature on the 2nd day preceding the onset [17], as well as acute coronary events and wind speed [18], and in particular excess number of wind speed increases [19], air temperature, and barometric pressure, on the day of the ambulance call and on the two previous days [18]. Associations have also been reported between out-of-hospital cardiac arrest of presumed cardiac etiology and wind speed, temperature, humidity, air pressure [20], paroxysmal atrial fibrillation and wind speed [21] as well as acute cerebrovascular events and wind speed [22, 23] or wind chill [23].

The foehn wind known as “halny”, occurring in southern Poland, has equally bad reputation as the foehn winds blowing in other parts of Europe and the world. The aim of this study was to examine the impact of the foehn wind and the related environmental variables on the incidence of cardiac events in the population inhabiting a region affected by halny wind. We also decided to determine whether the environmental changes coincide with or predate the event examined?

lines 93-96: The aim of this study was to examine the impact of the foehn wind and the related environmental variables on the incidence of cardiac events in the population inhabiting a region affected by halny wind. We also decided to determine whether the environmental changes coincide with or predate the event examined?

lines 124-132: 2.3. Data analyses

We examined the number of cardiac incidents occurring (i) on the days with halny wind blowing, and (ii) on days immediately preceding onset of halny. We compared these numbers to those occurring on the remaining days of the year. The statistical analyses were performed using Calc programme, by applying chi-square test of independence and test for two indicators of structure. Relationship between environmental variables and number of cardiac events was examined using multiple regression analysis. Distributions of mean daily air temperature, atmospheric humidity, speed of wind, pressure and total daily precipitation were shown in linear graphs. Statistical significance was assumed at p<0.05.

lines 186-235: 3. Results

3.4. Assessment of the association between environmental variables and number of cardiac events

The analyses did not confirm the regression model which assumed that the different environmental variables would impact the number of all cardiac events observed on all the days of the halny months (p=0.866, F=0.37). This regression model was only explained at a level of 11.0%.

Table 6. Assessment of the number of cardiac events/ number of admissions relative to the environmental variables on all days of the halny months.

Environmental variables vs number of cardiac events

Multiple regression

R2

Adjusted R2

F

p for regression model

b

Partial correlation

p

Mean air temperature (°C)

0.11

0.01

0.37

0.866

0.01

0.04

0.586

Mean atmospheric humidity (%)

-0.00

-0.01

0.873

Mean speed of wind (m/s)

-0.00

-0.04

0.613

Pressure (hPa)

-0.01

-0.07

0.413

Total daily precipitation (l/m3)

0.00

0.00

0,962

R2- regression model. Adjusted R2 – regression model disregards extreme values. F – result of Fisher’s test. b – regression coefficient. Partial correlation – the variable Xi is correlated with the variable Y after effect of all the other independent variables is taken into account (concurrent effect of all the four factors). p– significance level

Likewise, the analyses did not confirm the regression model for a shorter period of time (multiple regression analysis for 14 days of December), where it was assumed that different environmental variables would be related to the number of cardiac events observed (p=0.451, F=1.05). This regression model was explained at a level of 39.0%, and adjusted at a level of 2.0%.

Table 7. Assessment of the number of cardiac events/ number of admissions relative to the environmental variables from 1 to 14 December.

Environmental variables vs number of cardiac events

Multiple regression

R2

Adjusted R2

F

p for regression model

b

partial correlation

p

Mean air temperature (°C)

0.39

0.02

1.05

0.451

0.06

0.11

0.753

Mean atmospheric humidity (%)

0.05

0.38

0.283

Mean speed of wind (m/s)

0.68

0.45

0.197

Pressure (hPa)

0.00

0.01

0,970

Total daily precipitation (l/m3)

-0.11

-0.22

0.546

R2- regression model. Adjusted R2 – regression model disregards extreme values. F – result of Fisher’s test. b – regression coefficient. Partial correlation – the variable Xi is correlated with the variable Y after effect of all the other independent variables is taken into account (concurrent effect of all the four factors). p– significance level

3.5. Time series data showing daily levels of environmental variables and daily number of cardiac events in the period from 1 to 14 December 2011.

Table 8 presents distributions of mean daily air temperature, atmospheric humidity, speed of wind, pressure and total daily precipitation and daily number of cardiac events, in the period from 1 to 14 December.

Table 8. Example rows of time series data from the Tarnów dataset showing daily levels of environmental variables and daily number of cardiac events.

Date

Mean air temperature
(°C)

Mean atmospheric humidity
(%)

Mean speed of wind
(m/s)

Pressure

(hPa)

Total daily precipitation
(l/m3)

N cardiac events

1 Dec. 11

4.20

85.00

1.30

1026.80

0.00

2

2 Dec.11

4.00

73,90

2,90

1016.80

0.00

2

3 Dec.11

3.10

83.00

1.40

1014,90

0.20

1

4 Dec.11

8.10

71.10

2.60

1005.10

0.20

3

5 Dec.11

7.10

76.60

2.50

999.10

1,90

2

6 Dec.11

2.00

72.50

0,90

1009.10

0.00

0

7 Dec.11

2.40

80.30

2.50

1008.30

3,90

3

8 Dec.11

2,90

87.40

2,90

1011.00

4.40

2

9 Dec.11

2,90

71,90

2,90

1014.20

6.50

1

10 Dec.11

2.40

88.10

0,90

1013.80

0.30

1

11 Dec.11

0.20

80.30

0.80

1021.00

0.00

2

12 Dec.11

3.10

81,90

1.50

1016.10

1,90

1

13 Dec.11

4.70

92.60

1.10

1014.30

0.40

2

14 Dec.11

6.80

73.60

3.00

1012.40

0.00

2

Distributions of mean daily air temperature, atmospheric humidity, speed of wind, pressure and total daily precipitation and number of cardiac events were shown using linear graphs for the period from 1 to 14 December 2011 (Figure 1 and 2).

Figure 1. Distributions of mean daily air temperature, atmospheric humidity and speed of wind for the period from 1 to 14 December 2011.

Figure 2. Distributions of mean pressure, total daily precipitation and number of cardiac events for the period from 1 to 14 December 2011.

Analysis of the environmental variables (i.e. air temperature, atmospheric humidity, speed of wind, pressure and total daily precipitation) failed to demonstrate that any of these variables, or all of them jointly, affected the incidence of cardiac events on the days with halny or on the days preceding its onset.

lines 236-331: 4. Discussion

Based on a review of the literature it can be concluded that many researchers are investigating the association between wind-related environmental variables, and cardio- or cerebrovascular events, including the impact observed on days preceding onset of such weather conditions [14-23]. However, so far in Poland no such studies have been published yet. Boussoussou et al. have reported positive significant association between acute cardiovascular diseases and a cold front effect lagged by one day [14]. Mohammad et al. demonstrated that incidence of myocardial infarction increased with higher wind velocity, lower air temperature, lower atmospheric air pressure, and shorter sunshine duration [15]. Similarly, Honda et al. found that increased incidence of acute myocardial infarction corresponds to lower minimum temperature occurring on the 2nd day preceding the onset of wind [17]. Radišauskas et al. establish that myocardial infarction morbidity is directly correlated to wind speed, and atmospheric pressure and inversely correlated to atmospheric temperatures, and rainfall level [16]. Goerre et al. reported that the incidence of myocardial infarctions is significantly affected by wind activity, ambient pressure, and pressure gradient [15]. Vencloviene et al. showed that prognostic factors for the risk of acute coronary syndromes include such environmental variables as wind speed, daily air temperature, and barometric pressure on the day of the onset and on the two days preceding the ambulance call [18]. Likewise, Ezekowitz et al. showed relationships between the likelihood of an acute coronary event and rapid changes in weather conditions (including excessively frequent increases in wind speed) [19]. Hensel et al. showed that increasing wind speed corresponds to greater likelihood of out-of-hospital cardiac arrest of presumed cardiac aetiology [20]. Vencloviene et al. demonstrated that increased risk of paroxysmal atrial fibrillation occurring at night was linked with wind speed above the median (lag 2-4 days) [21]. Tamasauskiene et al. pointed out that a greater risk of ischemic stroke in older patients was associated with high wind speed on the day of the incident [22]. Similarly, Kim et al. showed that the odds ratios for wind speed, daily wind speed range, and wind chill for patients with ischemic stroke were slightly greater than in the total sample [23].

Effects of seasonal changes in environmental variables have also been pointed out. According to Gallerani et al., it is possible to observe 6-month-period regularities in the incidence of subarachnoid haemorrhage, spring and autumn being the peak seasons. The authors propose an explanation that during these transitional seasons the arterial wall is exposed to frequent stress episodes due to typical changes in temperature and that may result in aneurysmal rupture occurring more frequently than e.g. during the more stable periods of winter cold [26]. Investigators from the International Worldwide Registry reported that the high incidence of acute aortic dissection in winter is observed in various climate settings (both cold and temperate), which seems to suggest that the major mechanistic factors contributing to the phenomenon include the relative change in temperature, rather than absolute temperature, and/or circannual rhythms [27].

Contrary to the findings reported by other researchers [14-23], the current study did not confirm effects of environmental factors investigated, i.e. air temperature, atmospheric humidity, speed of wind, pressure and total daily precipitation in the incidence of cardiac events, either on halny days, or on days preceding the onset of halny.

Clinical studies provide evidence that some meteorological factors, and most importantly weather changes occurring rapidly, e.g. in one day, may play an important role in inducing subjectively perceived indisposition in healthy individuals and in promoting disease symptoms in patients. Notably, the same weather changes may induce completely different bodily responses. As an example, a large drop in barometric pressure will induce a decrease in blood pressure in some individuals and an increase in other people [2, 28-31]. Due to this no uniform recommendations for dealing with sudden changes in weather can be ad-dressed e.g. to all patients with cardiovascular problems.

The current study was an attempt to collect evidence related to the impact of foehn wind and the related environmental variables on the incidence of cardiac events. The basic meteotropic factor adopted in the study was defined as halny wind and the accompanying change from cold to warm weather front. The acquired results partly provide evidence related to effects of halny wind in the incidence of acute myocardial infarction. Although the findings covering the relevant period of time do not show significantly higher number of cardiac events on halny days, such correlation was identified in the case of days immediately preceding occurrence of halny. On the days preceding halny wind onset, incidence of cardiac events was higher than on the remaining days, however the findings showed no effect of the related environmental variables in the number of cardiac events observed during this time. Research conducted by other authors also suggests occurrence of similar synchronicities during a period ranging from 3 to 24 hours prior to an arrival of a weather front [32,33]. It has also been shown that meteorotropic responses in both healthy people and individuals affected by medical conditions may be observed more in advance of specific changes in weather; in extreme cases these may be “sensed” up to 48 hours before they happen. Such sensitivity may result from changes in electric field of the atmosphere, preceding arrival of a weather front [34]. It can be speculated that the greater number of cardiac incidents identified in our study 24 hours before onset of halny wind may be associated with the fact that indeed the wind is only an effect of earlier turbulent changes in atmospheric pressure systems. Hence, it is not halny wind itself that impacts cardiac events but the preceding changes in weather fronts. The factor of key importance here may be the drop in atmospheric pressure since onset of halny is determined by co-occurrence of high pressure on the southern side and low pressure on the northern side of the Tatra mountains. And this pressure drop may impact cardiac events. Other published studies confirm that changes in atmospheric pressure produce ad-verse effects in human body, possibly also leading to cardiac incidents [35-38].

As early as 1953, it was suggested by Petersen that human organism may be stimulated by specific relationships between weather fronts [6]. Cardiovascular problems are most frequently listed among meteorotropic diseases. In Poland the earliest studies focusing on impact of weather on functioning of cardio-vascular system were carried out in the mid-1800s [6]. Research shows impact of short-term weather changes on intensity of ischemic heart disease symptoms, increased oscillations in arterial blood pressure, and higher incidence of acute myocardial infarction. Over 34% of fatal cases of myocardial infarction co-occur with a passage of weather fronts, compared to only 6% observed during periods of calm weather linked with high pressure [39]. A survey conducted by Baranowska showed that in the respondents’ opinion, meteoropaths are ad-versely affected by weather transitions from highs to lows [6]. In a study conducted by Höppe, individuals sensitive to weather, living in northern Germany, negatively assessed stormy weather (30%) and influx of cold air (29%). On the other hand, inhabitants of south-ern Germany experience mood decline accompanying influx of warm (30%) rather than cold air (18%). This difference may be associated with the quite frequent foehn winds in southern Germany [40]. Mackensen et al. investigated effects of weather in the sense of well-being observed in people living in Germany and in Canada. The authors reported weather sensitivity in 54.5% of the subjects representing German population while in Canada the rate amounted to 61% [12]. Likewise, Shiue et al. found a relationship between weather conditions, including wind speed, and hospital admissions due to heart disease in Germany [41]. The same relationship was shown by studies conducted in Switzerland [42] and Lithuania [16].

It is difficult to clearly establish associations between meteorological or synoptic factors and decline in people’s mood or well-being. The reasons for that may lie e.g. in varied responses to weather related stimuli, or in synergistic effects produced by various meteorological factors. Therefore, results obtained by various researchers frequently cannot be compared due to the fact that different methodologies are applied in studies.

Lines 346-357: 6. Conclusions

  1. On the days with halny wind, incidence of cardiac events is similar to that on the remaining days of the year.
  2. The findings show no effect of the related environmental variables on the number of cardiac events observed.
  3. During 24 hours preceding onset of halny wind, incidence of acute myocardial infarction was significantly higher than during the remaining days of the year, however no evidence was found to confirm impact of the related environmental variables on the number of admissions during the relevant period.
  4. To establish a cause-and-effect relationship, more comprehensive data should be acquired from a larger number of regions affected by halny wind in Poland, and the relevant observations should be continued for a period longer than one calendar year.

Point 1: Minor remarks:

p3, line 99-101: The number of male patients is observed, so do the authors mean that this was “confirmed in the test for significance”?

p5 line 141: This idea that some people are sensitive to weather changes (meteoropaths) seems important and should be cited in the introduction, not in the discussion.  The lack of supporting literature completely undermines the objective of this manuscript.

Response 2: Thank you for the helpful suggestions. Yes, the statistical analysis confirmed that, compared to the number of females, significantly more male patients were taken into account in our study. The relevant fragment has been edited to make it clear. In accordance with the Reviewer’s suggestion, we have moved the relevant information from the discussion to introduction and we have used supporting references.

BEFORE

lines 99-101: The group included 168 females (36.1%) and 297 males (63.9%). The number of male patients admitted was significantly higher compared to the female patients, which was confirmed in the test for significance of structure indicators (p<0.001)

lines 140-151: 4. Discussion

People with high sensitivity to weather changes are referred to as meteoropaths. They present increased tension of the vegetative system and poor adaptative responses. Effects of weather fronts, in particular cold fronts, include decrease in the temperature and increase in the atmospheric pressure, responses to which include vasoconstriction, potentially posing a burden for the circulatory system [11-13] Nowadays, it is estimated that the number of me-teoropaths is consistently growing. They constitute approximately 50%-70% of the healthy populations. It is also believed that meteoropathy is a “disease” of affluence, caused by stress, work overload, and unhealthy lifestyles, since meteoropaths mainly include residents of urban areas, whose immunity is affected by the long time spent in the artificial microcli-mates of their homes and offices, by a lack of exercise and by polluted air. It is also pointed out that meteoropathy is most commonly observed in individuals affected by coronary dis-ease, hypertension and ulcer disease [11-14].

AFTER

lines 144-146: The group included 168 females (36.1%) and 297 males (63.9%). The number of male patients admitted was significantly higher compared to the female patients. This fact was confirmed in the test for significance (p<0.001)

BEFORE

lines 67-96: 1. Introduction

People with high sensitivity to weather changes are referred to as meteoropaths. They present increased tension of the vegetative system and poor adaptative responses. Effects of weather fronts, in particular cold fronts, include decrease in the temperature and increase in the atmospheric pressure, responses to which include vasoconstriction, potentially posing a burden for the circulatory system [8-10] Nowadays, it is estimated that the number of meteoropaths is consistently growing. According to the data from the Institute of Meteorology and Water Management, as many as 50-70% of Poles respond to weather changes [11]. Similar statistics have been published in other countries, e.g. in Germany 54.5% of the population is weather sensitive, in Canada the relevant rate amounts to 61% [12]. It is also believed that meteoropathy is a “disease” of affluence, caused by stress, work overload, and unhealthy lifestyles, since meteoropaths mainly include residents of urban areas, whose immunity is affected by the long time spent in the artificial microclimates of their homes and offices, by a lack of exercise and by polluted air. It has also been pointed out that meteoropathy is most commonly observed in individuals affected by coronary disease, hypertension and ulcer disease [8-10,13].

A large body of evidence reported in the literature shows there is an association between wind- related environmental variables, and cardiovascular or cerebrovascular events, with impact observed also on the preceding days [14-23]. The related research findings focus on acute cardiovascular diseases and atmospheric front effect lagged by one day [14], myocardial infarction in relation to wind velocity, atmospheric air pressure, air temperature [15, 16] and lower minimum temperature on the 2nd day preceding the onset [17], as well as acute coronary events and wind speed [18], and in particular excess number of wind speed increases [19], air temperature, and barometric pressure, on the day of the ambulance call and on the two previous days [18]. Associations have also been reported between out-of-hospital cardiac arrest of presumed cardiac etiology and wind speed, temperature, humidity, air pressure [20], paroxysmal atrial fibrillation and wind speed [21] as well as acute cerebrovascular events and wind speed [22, 23] or wind chill [23].

The aim of this study was to examine the impact of the foehn wind and the related environmental variables on the incidence of cardiac events in the population inhabiting a region affected by halny wind. We also decided to determine whether the environmental changes coincide with or predate the event examined?

Round 2

Reviewer 3 Report

I appreciate the authors have put in considerable effort in this revised manuscript.  The motivation is a bit clearer, albeit largely provided in the Discussion rather than the Introduction.  However, the scientific merit is still lacking.

Despite the revision to the Discussion, this section continues to be largely an extension of the literature review rather than a discussion of the results presented by the authors. Lines 241 to 271, lines 276 to 283, and lines 308 to 331 should be in the introduction.  Especially lines 327 to 331 should appear before the current study is introduced as it provides some of the motivation.  Similarly, most of the text in lines 293 to 307 could be rephrased and moved to the introduction, and the results of the current study could be compared to the results of these other published studies succinctly.  The Discussion section should be discussing the results – what they mean (inference), and why they are important or useful.  Not one of the 8 tables or 2 figures is discussed.  As a naïve reader looking at these tables and figures, I have to wonder “So what?”  For example, looking at the last subplot in Figure 2, which shows the number of cardiac events over time and their relationship to days with halny winds, I wonder what important piece of information this plot is showing. Should I find it interesting that the two days with the highest cases occur before a halny day?  Is there some temporal pattern?  This is the sort of content that should be in the Discussion.

Even if the authors improve the Introduction and Discussion section, the methodology is not appropriate for achieving the paper’s aims, namely to examine the impact of halny winds on cardiac event incidence, either directly or indirectly through changes in the environment.  This is for two main reasons.  First, the authors obtained data relating to 465 cardiology patients covering a 365 day period, but the halny days and number of admissions on these days or days either side of halny days only account for a small portion of the total data.  This severely reduces the statistical power.  It is not clear how large the “southern Poland” study region is, but if it is quite large, then sourcing records from other hospitals in the region should be considered.  Similarly, additional years of data would be helpful to increase these numbers and further determine what role temporal effects play.  At the very least, these issues should be mentioned as limitations of the study.  Second, the main statistical test is a chi-square test of independence. This seems inappropriate since 1) all the literature, according to the authors, provides evidence that these variables are indeed related (or correlated), and 2) such a test doesn’t address the aim of the paper.  I reiterate my comment on the previous version of this manuscript, that to answer the aim of the paper, the authors need to answer HOW the variables are correlated.  I suggest a longitudinal study which will help identify and characterise temporal autocorrelation between admissions and weather events caused by halny.

Minor remarks:

Line 96 shouldn’t have a question mark.

The x-axis legends in most of the plots seem incorrectly formatted (i.e. do not look like dates in the month of December).

Line 217: I don’t think “distributions” is the right word.

Author Response

Response to Reviewer 3 Comments

Dear Reviewer,        

Thank you for re-reviewing our article titled “Impact of foehn wind and related environmental variables on the incidence of cardiac events”. We have made every effort to improve our manuscript, as guided by the reviewer’s helpful suggestions.

We thank the reviewer of all the comments. Answers are summarised below. All changes are highlighted as red text in the manuscript.

We hope you will be pleased with the changes, and support the publication of our revised manuscript.

With kind regards,

The authors of the article

Point 1: Major remarks:

I appreciate the authors have put in considerable effort in this revised manuscript.  The motivation is a bit clearer, albeit largely provided in the Discussion rather than the Introduction.  However, the scientific merit is still lacking.

Despite the revision to the Discussion, this section continues to be largely an extension of the literature review rather than a discussion of the results presented by the authors. Lines 241 to 271, lines 276 to 283, and lines 308 to 331 should be in the introduction.  Especially lines 327 to 331 should appear before the current study is introduced as it provides some of the motivation.  Similarly, most of the text in lines 293 to 307 could be rephrased and moved to the introduction, and the results of the current study could be compared to the results of these other published studies succinctly.  The Discussion section should be discussing the results – what they mean (inference), and why they are important or useful.  Not one of the 8 tables or 2 figures is discussed.  As a naïve reader looking at these tables and figures, I have to wonder “So what?”  For example, looking at the last subplot in Figure 2, which shows the number of cardiac events over time and their relationship to days with halny winds, I wonder what important piece of information this plot is showing. Should I find it interesting that the two days with the highest cases occur before a halny day?  Is there some temporal pattern?  This is the sort of content that should be in the Discussion.

Even if the authors improve the Introduction and Discussion section, the methodology is not appropriate for achieving the paper’s aims, namely to examine the impact of halny winds on cardiac event incidence, either directly or indirectly through changes in the environment.  This is for two main reasons.  First, the authors obtained data relating to 465 cardiology patients covering a 365 day period, but the halny days and number of admissions on these days or days either side of halny days only account for a small portion of the total data.  This severely reduces the statistical power.  It is not clear how large the “southern Poland” study region is, but if it is quite large, then sourcing records from other hospitals in the region should be considered.  Similarly, additional years of data would be helpful to increase these numbers and further determine what role temporal effects play.  At the very least, these issues should be mentioned as limitations of the study.  Second, the main statistical test is a chi-square test of independence. This seems inappropriate since 1) all the literature, according to the authors, provides evidence that these variables are indeed related (or correlated), and 2) such a test doesn’t address the aim of the paper.  I reiterate my comment on the previous version of this manuscript, that to answer the aim of the paper, the authors need to answer HOW the variables are correlated.  I suggest a longitudinal study which will help identify and characterise temporal autocorrelation between admissions and weather events caused by halny.

Response 1: Thank you for this valuable comment. In accordance with the Reviewer’s suggestion, we have revised the Introduction and the Discussion (we have discussed the results, tables and figures) and we have added the study limitations, and changed the Conclusions. To answer the aim of the paper, we have added the statistics describing temporal correlations between admissions and weather events caused by halny which, consequently, allowed us to answer how the variables are correlated. 

We apologise but in the previous version of the manuscript we overlooked a detailed justification for the applied tests. Chi-squared test was used to compare the frequency of the specific phenomenon occurring relative to the patients’ sex, in order to confirm or refute the effect of this variable. The test was used because of the qualitative nature of the data (Table 1 and 3). On the other hand, in Tables 4 and 5, a two-tailed test for significance of structure indicators was applied (to assess frequency of one phenomenon coinciding with another phenomenon).

To identify and characterise temporal autocorrelation between admissions and weather events caused by halny we applied Pearson’s correlation coefficient in selected periods of time (days with halny, days preceding halny onset, total days with and preceding halny, other days, all the days combined).

Additional relationship between environmental variables and number of cardiac events was examined using multiple regression analysis:

- Assessment of the number of admissions relative to the environmental variables, throughout the year in the consecutive months from January 2011 to December 2011

- Assessment of the number of cardiac events/ number of admissions relative to the environmental variables on all days of the halny months (January, March, October, November, December)

- Assessment of the number of cardiac events/ number of admissions relative to the environmental variables from 1 to 14 December (we selected December because halny wind occurred most frequently during this month, i.e. in 5-6 Dec, 8-9 Dec, 12-17 Dec)

We also took into account time series data showing daily levels of environmental variables and daily number of cardiac events in the period from 1 to 14 December 2011.

BEFORE

Lines 33-96:

  1. Introduction

Cardiovascular disorders are the most common cause of mortality worldwide, including most European countries, and Poland [1]. Seasonal peaks in acute myocardial infarction incidence have been widely reported. Weather has been postulated to be one of the elements greatly contributing to this association [2].

The term “föhn” is used with reference to a type of strong and turbulent wind blowing down from mountain crests towards the lee side, and bringing dry and warm air to the surface. Foehn events rather frequently occur in the Alps, or more specifically on the northern side of the mountains, and constitute a significant challenge in weather forecasting for the Alpine valleys, inhabited by a lot of people. The northern Alpine foehn is frequently observed in the prefrontal environment. The foehn clearance reflects the leeward range of the foehn in the mountains, i.e. a region with no clouds where dry and warm air occurs at the surface, just as in the foehn region but with no strong winds. This meteorological phenomenon is not unique to the Alps, and it can be observed over other high mountains worldwide [3].

A local variety of foehn, “halny” wind is produced by violent flow of air masses from the southern side of the Carpathians or the Sudetes. This happens when Poland is approached from the west by a deep cyclone (low-pressure area), while the centre of anticyclone (high-pressure) is in Ukraine [3,4]. In addition to an abrupt change in temperature and large speed of the wind, there are also other consequences of halny wind. The first one involves rapid pressure drop, and the other one is linked with increased condensation of positive ions in the air as a result of which microscopic particles of gases and aerosols, including harmful elements and chemical compounds, can easily penetrate organisms [3,4]. Halny wind frequently leads to considerable increase in the temperature and decrease in humidity on the leeward side of the mountains. The strong abrupt gusts of wind at times reach a speed typical for hurricane, and are accompanied with characteristic calm intervals. The foehn winds are observed in the Tatras rather frequently, particularly during the cold seasons. The criteria commonly applied by Polish researchers investigating foehn winds occurring in the mountainous areas include: direction (usually 90-270) and wind speed (> 4 m/s, commonly ≥10 m/s, at times increasing to even 60 m/s), occurrence of strong gusts and calm intervals, sudden rise in the air temperature (by several degrees in a short time) and decrease in the humidity (commonly ≤70%), changes in atmospheric pressure and / or certain types of the clouds. In foothill and lowland areas, the wind is dry and warm [4].

It is believed that foehn wind may lead to increased aggressiveness, more severe depressive states, emotional crises ending with suicide attempts, agitation, anxiety, fatigue, weariness, muscle and joint pain, problems with concentration, sleep disorders, migraine, as well as headaches [4-7]. 

People with high sensitivity to weather changes are referred to as meteoropaths. They present increased tension of the vegetative system and poor adaptative responses. Effects of weather fronts, in particular cold fronts, include decrease in the temperature and increase in the atmospheric pressure, responses to which include vasoconstriction, potentially posing a burden for the circulatory system [8-10] Nowadays, it is estimated that the number of meteoropaths is consistently growing. According to the data from the Institute of Meteorology and Water Management, as many as 50-70% of Poles respond to weather changes [11]. Similar statistics have been published in other countries, e.g. in Germany 54.5% of the population is weather sensitive, in Canada the relevant rate amounts to 61% [12]. It is also believed that meteoropathy is a “disease” of affluence, caused by stress, work overload, and unhealthy lifestyles, since meteoropaths mainly include residents of urban areas, whose immunity is affected by the long time spent in the artificial microclimates of their homes and offices, by a lack of exercise and by polluted air. It has also been pointed out that meteoropathy is most commonly observed in individuals affected by coronary disease, hypertension and ulcer disease [8-10,13].

A large body of evidence reported in the literature shows there is an association between wind- related environmental variables, and cardiovascular or cerebrovascular events, with impact observed also on the preceding days [14-23]. The related research findings focus on acute cardiovascular diseases and atmospheric front effect lagged by one day [14], myocardial infarction in relation to wind velocity, atmospheric air pressure, air temperature [15, 16] and lower minimum temperature on the 2nd day preceding the onset [17], as well as acute coronary events and wind speed [18], and in particular excess number of wind speed increases [19], air temperature, and barometric pressure, on the day of the ambulance call and on the two previous days [18]. Associations have also been reported between out-of-hospital cardiac arrest of presumed cardiac etiology and wind speed, temperature, humidity, air pressure [20], paroxysmal atrial fibrillation and wind speed [21] as well as acute cerebrovascular events and wind speed [22, 23] or wind chill [23].

The foehn wind known as “halny”, occurring in southern Poland, has equally bad reputation as the foehn winds blowing in other parts of Europe and the world. The aim of this study was to examine the impact of the foehn wind and the related environmental variables on the incidence of cardiac events in the population inhabiting a region affected by halny wind. We also decided to determine whether the environmental changes coincide with or predate the event examined?

Lines 124-132:

2.3. Data analyses

We examined the number of cardiac incidents occurring (i) on the days with halny wind blowing, and (ii) on days immediately preceding onset of halny. We compared these numbers to those occurring on the remaining days of the year. The statistical analyses were performed using Calc programme, by applying chi-square test of independence and test for two indicators of structure. Relationship between environmental variables and number of cardiac events was examined using multiple regression analysis. Distributions of mean daily air temperature, atmospheric humidity, speed of wind, pressure and total daily precipitation were shown in linear graphs. Statistical significance was assumed at p<0.05.

Lines 236-345:

  1. Discussion

Based on a review of the literature it can be concluded that many researchers are investigating the association between wind-related environmental variables, and cardio- or cerebrovascular events, including the impact observed on days preceding onset of such weather conditions [14-23]. However, so far in Poland no such studies have been published yet. Boussoussou et al. have reported positive significant association between acute cardiovascular diseases and a cold front effect lagged by one day [14]. Mohammad et al. demonstrated that incidence of myocardial infarction increased with higher wind velocity, lower air temperature, lower atmospheric air pressure, and shorter sunshine duration [15]. Similarly, Honda et al. found that increased incidence of acute myocardial infarction corresponds to lower minimum temperature occurring on the 2nd day preceding the onset of wind [17]. Radišauskas et al. establish that myocardial infarction morbidity is directly correlated to wind speed, and atmospheric pressure and inversely correlated to atmospheric temperatures, and rainfall level [16]. Goerre et al. reported that the incidence of myocardial infarctions is significantly affected by wind activity, ambient pressure, and pressure gradient [15]. Vencloviene et al. showed that prognostic factors for the risk of acute coronary syndromes include such environmental variables as wind speed, daily air temperature, and barometric pressure on the day of the onset and on the two days preceding the ambulance call [18]. Likewise, Ezekowitz et al. showed relationships between the likelihood of an acute coronary event and rapid changes in weather conditions (including excessively frequent increases in wind speed) [19]. Hensel et al. showed that increasing wind speed corresponds to greater likelihood of out-of-hospital cardiac arrest of presumed cardiac aetiology [20]. Vencloviene et al. demonstrated that increased risk of paroxysmal atrial fibrillation occurring at night was linked with wind speed above the median (lag 2-4 days) [21]. Tamasauskiene et al. pointed out that a greater risk of ischemic stroke in older patients was associated with high wind speed on the day of the incident [22]. Similarly, Kim et al. showed that the odds ratios for wind speed, daily wind speed range, and wind chill for patients with ischemic stroke were slightly greater than in the total sample [23].

Effects of seasonal changes in environmental variables have also been pointed out. According to Gallerani et al., it is possible to observe 6-month-period regularities in the incidence of subarachnoid haemorrhage, spring and autumn being the peak seasons. The authors propose an explanation that during these transitional seasons the arterial wall is exposed to frequent stress episodes due to typical changes in temperature and that may result in aneurysmal rupture occurring more frequently than e.g. during the more stable periods of winter cold [26]. Investigators from the International Worldwide Registry reported that the high incidence of acute aortic dissection in winter is observed in various climate settings (both cold and temperate), which seems to suggest that the major mechanistic factors contributing to the phenomenon include the relative change in temperature, rather than absolute temperature, and/or circannual rhythms [27].

Contrary to the findings reported by other researchers [14-23], the current study did not confirm effects of environmental factors investigated, i.e. air temperature, atmospheric humidity, speed of wind, pressure and total daily precipitation in the incidence of cardiac events, either on halny days, or on days preceding the onset of halny.

Clinical studies provide evidence that some meteorological factors, and most importantly weather changes occurring rapidly, e.g. in one day, may play an important role in inducing subjectively perceived indisposition in healthy individuals and in promoting disease symptoms in patients. Notably, the same weather changes may induce completely different bodily responses. As an example, a large drop in barometric pressure will induce a decrease in blood pressure in some individuals and an increase in other people [2, 28-31]. Due to this no uniform recommendations for dealing with sudden changes in weather can be ad-dressed e.g. to all patients with cardiovascular problems.

The current study was an attempt to collect evidence related to the impact of foehn wind and the related environmental variables on the incidence of cardiac events. The basic meteotropic factor adopted in the study was defined as halny wind and the accompanying change from cold to warm weather front. The acquired results partly provide evidence related to effects of halny wind in the incidence of acute myocardial infarction. Although the findings covering the relevant period of time do not show significantly higher number of cardiac events on halny days, such correlation was identified in the case of days immediately preceding occurrence of halny. On the days preceding halny wind onset, incidence of cardiac events was higher than on the remaining days, however the findings showed no effect of the related environmental variables in the number of cardiac events observed during this time. Research conducted by other authors also suggests occurrence of similar synchronicities during a period ranging from 3 to 24 hours prior to an arrival of a weather front [32,33]. It has also been shown that meteorotropic responses in both healthy people and individuals affected by medical conditions may be observed more in advance of specific changes in weather; in extreme cases these may be “sensed” up to 48 hours before they happen. Such sensitivity may result from changes in electric field of the atmosphere, preceding arrival of a weather front [34]. It can be speculated that the greater number of cardiac incidents identified in our study 24 hours before onset of halny wind may be associated with the fact that indeed the wind is only an effect of earlier turbulent changes in atmospheric pressure systems. Hence, it is not halny wind itself that impacts cardiac events but the preceding changes in weather fronts. The factor of key importance here may be the drop in atmospheric pressure since onset of halny is determined by co-occurrence of high pressure on the southern side and low pressure on the northern side of the Tatra mountains. And this pressure drop may impact cardiac events. Other published studies confirm that changes in atmospheric pressure produce ad-verse effects in human body, possibly also leading to cardiac incidents [35-38].

As early as 1953, it was suggested by Petersen that human organism may be stimulated by specific relationships between weather fronts [6]. Cardiovascular problems are most frequently listed among meteorotropic diseases. In Poland the earliest studies focusing on impact of weather on functioning of cardio-vascular system were carried out in the mid-1800s [6]. Research shows impact of short-term weather changes on intensity of ischemic heart disease symptoms, increased oscillations in arterial blood pressure, and higher incidence of acute myocardial infarction. Over 34% of fatal cases of myocardial infarction co-occur with a passage of weather fronts, compared to only 6% observed during periods of calm weather linked with high pressure [39]. A survey conducted by Baranowska showed that in the respondents’ opinion, meteoropaths are ad-versely affected by weather transitions from highs to lows [6]. In a study conducted by Höppe, individuals sensitive to weather, living in northern Germany, negatively assessed stormy weather (30%) and influx of cold air (29%). On the other hand, inhabitants of south-ern Germany experience mood decline accompanying influx of warm (30%) rather than cold air (18%). This difference may be associated with the quite frequent foehn winds in southern Germany [40]. Mackensen et al. investigated effects of weather in the sense of well-being observed in people living in Germany and in Canada. The authors reported weather sensitivity in 54.5% of the subjects representing German population while in Canada the rate amounted to 61% [12]. Likewise, Shiue et al. found a relationship between weather conditions, including wind speed, and hospital admissions due to heart disease in Germany [41]. The same relationship was shown by studies conducted in Switzerland [42] and Lithuania [16].

It is difficult to clearly establish associations between meteorological or synoptic factors and decline in people’s mood or well-being. The reasons for that may lie e.g. in varied responses to weather related stimuli, or in synergistic effects produced by various meteorological factors. Therefore, results obtained by various researchers frequently cannot be compared due to the fact that different methodologies are applied in studies.

  1. Strengths and limitations of this study

This is the first exploratory article, the aim of which is to provoke discussion on the impact of foehn wind and the related environmental variables on the incidence of cardiac events in the population inhabiting a region affected by halny wind in Poland. Although the findings covering the relevant period of time do not show significantly higher number of cardiac events on halny days, such correlation was identified in the case of days immediately preceding the related change in weather. On days preceding halny wind onset, incidence of cardiac events was higher than on the remaining days, however the related environmental variables were not found to affect the incidence of cardiac events. To establish a cause-and-effect relationship the relevant data should be acquired from more regions affected by halny wind in Poland. The second limitation of the study is the insufficient duration of the observation period, and the fact that the analyses took into account data related to one calendar year only.

Lines 346-357:

Conclusion

1.On the days with halny wind, incidence of cardiac events is similar to that on the remaining days of the year.

2. The findings show no effect of the related environmental variables on the number of cardiac events observed.

3. During 24 hours preceding onset of halny wind, incidence of acute myocardial infarction was significantly higher than during the remaining days of the year, however no evidence was found to confirm impact of the related environmental variables on the number of admissions during the relevant period.

4. To establish a cause-and-effect relationship, more comprehensive data should be acquired from a larger number of regions affected by halny wind in Poland, and the relevant observations should be continued for a period longer than one calendar year.

AFTER

Lines 33-159:

  1. Introduction

Cardiovascular disorders are the most common cause of mortality worldwide, including most European countries, and Poland [1]. Seasonal peaks in acute myocardial infarction incidence have been widely reported. Weather has been postulated to be one of the elements greatly contributing to this association [2]. However, it is difficult to clearly establish relationships between meteorological or synoptic factors and decline in people’s mood or well-being. The reasons for that may lie e.g. in varied responses to weather related stimuli, or in synergistic effects produced by various meteorological factors. Due to this, results obtained by various researchers frequently cannot be compared because of the different methodologies applied in studies. Nevertheless, in order to get better understanding of health-related effects produced by various environmental factors, there is a need for further research focusing on possible links between specific medical conditions and selected weather phenomena.

The term “föhn” is used with reference to a type of strong and turbulent wind blowing down from mountain crests towards the lee side, and bringing dry and warm air to the surface. Foehn events rather frequently occur in the Alps, or more specifically on the northern side of the mountains, and constitute a significant challenge in weather forecasting for the Alpine valleys, inhabited by a lot of people. The northern Alpine foehn is frequently observed in the prefrontal environment. The foehn clearance reflects the leeward range of the foehn in the mountains, i.e. a region with no clouds where dry and warm air occurs at the surface, just as in the foehn region but with no strong winds. This meteorological phenomenon is not unique to the Alps, and it can be observed over other high mountains worldwide [3].

A local variety of foehn, “halny” wind is produced by violent flow of air masses from the southern side of the Carpathians or the Sudetes. This happens when Poland is approached from the west by a deep cyclone (low-pressure area), while the centre of anticyclone (high-pressure) is in Ukraine [3,4]. In addition to an abrupt change in temperature and large speed of the wind, there are also other consequences of halny wind. The first one involves rapid pressure drop, and the other one is linked with increased condensation of positive ions in the air as a result of which microscopic particles of gases and aerosols, including harmful elements and chemical compounds, can easily penetrate organisms [3,4]. Halny wind frequently leads to considerable increase in the temperature and decrease in humidity on the leeward side of the mountains. The strong abrupt gusts of wind at times reach a speed typical for hurricane, and are accompanied with characteristic calm intervals. The foehn winds are observed in the Tatras rather frequently, particularly during the cold seasons. The criteria commonly applied by Polish researchers investigating foehn winds occurring in the mountainous areas include: direction (usually 90-270) and wind speed (> 4 m/s, commonly ≥10 m/s, at times increasing to even 60 m/s), occurrence of strong gusts and calm intervals, sudden rise in the air temperature (by several degrees in a short time) and decrease in the humidity (commonly ≤70%), changes in atmospheric pressure and / or certain types of the clouds. In foothill and lowland areas, the wind is dry and warm [4].

It is believed that foehn wind may lead to increased aggressiveness, more severe depressive states, emotional crises ending with suicide attempts, agitation, anxiety, fatigue, weariness, muscle and joint pain, problems with concentration, sleep disorders, migraine, as well as headaches [4-7]. 

People with high sensitivity to weather changes are referred to as meteoropaths. They present increased tension of the vegetative system and poor adaptative responses. Effects of weather fronts, in particular cold fronts, include decrease in the temperature and increase in the atmospheric pressure, responses to which include vasoconstriction, potentially posing a burden for the circulatory system [8-10] Nowadays, it is estimated that the number of meteoropaths is consistently growing. According to the data from the Institute of Meteorology and Water Management, as many as 50-70% of Poles respond to weather changes [11]. Similar statistics have been published in other countries, e.g. in Germany 54.5% of the population is weather sensitive, in Canada the relevant rate amounts to 61% [12]. It is also believed that meteoropathy is a “disease” of affluence, caused by stress, work overload, and unhealthy lifestyles, since meteoropaths mainly include residents of urban areas, whose immunity is affected by the long time spent in the artificial microclimates of their homes and offices, by a lack of exercise and by polluted air. It has also been pointed out that meteoropathy is most commonly observed in individuals affected by coronary disease, hypertension and ulcer disease [8-10,13].

A large body of evidence reported in the literature shows there is an association between wind- related environmental variables, and cardiovascular or cerebrovascular events, with impact observed also on the preceding days [14-23]. Boussoussou et al. have reported positive significant association between acute cardiovascular diseases and a cold front effect lagged by one day [14]. Mohammad et al. demonstrated that incidence of myocardial infarction increased with higher wind velocity, lower air temperature, lower atmospheric air pressure, and shorter sunshine duration [15]. Similarly, Honda et al. found that increased incidence of acute myocardial infarction corresponds to lower minimum temperature occurring on the 2nd day preceding the onset of wind [17]. Radišauskas et al. establish that myocardial infarction morbidity is directly correlated to wind speed, and atmospheric pressure and inversely correlated to atmospheric temperatures, and rainfall level [16]. Goerre et al. reported that the incidence of myocardial infarctions is significantly affected by wind activity, ambient pressure, and pressure gradient [15]. Vencloviene et al. showed that prognostic factors for the risk of acute coronary syndromes include such environmental variables as wind speed, daily air temperature, and barometric pressure on the day of the onset and on the two days preceding the ambulance call [18]. Likewise, Ezekowitz et al. showed relationships between the likelihood of an acute coronary event and rapid changes in weather conditions (including excessively frequent increases in wind speed) [19]. Hensel et al. showed that increasing wind speed corresponds to greater likelihood of out-of-hospital cardiac arrest of presumed cardiac aetiology [20]. Vencloviene et al. demonstrated that increased risk of paroxysmal atrial fibrillation occurring at night was linked with wind speed above the median (lag 2-4 days) [21]. Tamasauskiene et al. pointed out that a greater risk of ischemic stroke in older patients was associated with high wind speed on the day of the incident [22]. Similarly, Kim et al. showed that the odds ratios for wind speed, daily wind speed range, and wind chill for patients with ischemic stroke were slightly greater than in the total sample [23].

Effects of seasonal changes in environmental variables have also been pointed out. According to Gallerani et al., it is possible to observe 6-month-period regularities in the incidence of subarachnoid haemorrhage, spring and autumn being the peak seasons. The authors propose an explanation that during these transitional seasons the arterial wall is exposed to frequent stress episodes due to typical changes in temperature and that may result in aneurysmal rupture occurring more frequently than e.g. during the more stable periods of winter cold [24]. Investigators from the International Worldwide Registry reported that the high incidence of acute aortic dissection in winter is observed in various climate settings (both cold and temperate), which seems to suggest that the major mechanistic factors contributing to the phenomenon include the relative change in temperature, rather than absolute temperature, and/or circannual rhythms [25].

Clinical studies provide evidence that some meteorological factors, and most importantly weather changes occurring rapidly, e.g. in one day, may play an important role in inducing subjectively perceived indisposition in healthy individuals and in promoting disease symptoms in patients. Notably, the same weather changes may induce completely different bodily responses. As an example, a large drop in barometric pressure will induce a decrease in blood pressure in some individuals and an increase in other people [2, 26-29]. Due to this no uniform recommendations for dealing with sudden changes in weather can be ad-dressed e.g. to all patients with cardiovascular problems.

It has been shown that meteorotropic responses in both healthy people and individuals affected by medical conditions may be observed more in advance of specific changes in weather; in extreme cases these may be “sensed” up to 48 hours before they actually happen. Such sensitivity may result from the changes in the electric field of the atmosphere, preceding arrival of a weather front [30-32]. It has also been reported that changes in atmospheric pressure produce adverse effects in human body, possibly also leading to cardiac incidents [33-36].

As early as 1953, it was suggested by Petersen that human organism may be stimulated by specific relationships between weather fronts [6]. Cardiovascular problems are most frequently listed among meteorotropic diseases. In Poland the earliest studies focusing on impact of weather on functioning of cardio-vascular system were carried out in the mid-1800s [6]. Research shows impact of short-term weather changes on intensity of ischemic heart disease symptoms, increased oscillations in arterial blood pressure, and higher incidence of acute myocardial infarction. Over 34% of fatal cases of myocardial infarction co-occur with a passage of weather fronts, compared to only 6% observed during periods of calm weather linked with high pressure [37]. A survey conducted by Baranowska showed that in the respondents’ opinion, meteoropaths are ad-versely affected by weather transitions from highs to lows [6]. In a study conducted by Höppe, individuals sensitive to weather, living in northern Germany, negatively assessed stormy weather (30%) and influx of cold air (29%). On the other hand, inhabitants of south-ern Germany experience mood decline accompanying influx of warm (30%) rather than cold air (18%). This difference may be associated with the quite frequent foehn winds in southern Germany [38]. Mackensen et al. investigated effects of weather in the sense of well-being observed in people living in Germany and in Canada. The authors reported weather sensitivity in 54.5% of the subjects representing German population while in Canada the rate amounted to 61% [12]. Likewise, Shiue et al. found a relationship between weather conditions, including wind speed, and hospital admissions due to heart disease in Germany [39]. The same relationship was shown by studies conducted in Switzerland [40] and Lithuania [16].

The foehn wind known as “halny”, occurring in southern Poland, has equally bad reputation as the foehn winds blowing in other parts of Europe and the world. The aim of this study was to examine the impact of the foehn wind and the related environmental variables on the incidence of cardiac events in the population inhabiting a region affected by halny wind. We also decided to determine whether the environmental changes coincide with or predate the event examined.

Lines 185-198:

2.3. Data analyses

We examined the number of cardiac incidents occurring (i) on the days with halny wind blowing, and (ii) on days immediately preceding onset of halny. We compared these numbers to those occurring on the remaining days of the year. The statistical analyses were performed using Calc programme. Chi-squared test was applied to compare the frequency of the specific phenomenon occurring relative to the patients’ sex, in order to confirm or refute the effect of this variable. The test was used because of the qualitative nature of the data. Frequency of one phenomenon coinciding with another phenomenon was examined with two-tailed test for significance of structure indicators. Temporal correlations between admissions and weather events caused by halny were analysed using Pearson’s correlation coefficient. Additional relationship between environmental variables and number of cardiac events was examined using multiple regression analysis. Changes in mean daily air temperature, atmospheric humidity, speed of wind, pressure and total daily precipitation were shown in linear graphs. Statistical significance was assumed at p<0.05.

Lines 250-256:

3.4. Temporal correlation between admissions and weather events caused by halny

No statistically significant correlations were found between the environmental variables and number of cardiac events in any of the periods investigated. The analyses took into account a period of five months during which halny wind occurred (Table 6).

Table 6. Relationship between the number of cardiac events and environmental variables in the selected periods of time.

Halny days

Days preceding halny

Total number of days with halny and preceding halny

Other days

Total number of days

Mean air temperature (°C)

Mean atmospheric humidity (%)

0.33

0.21

0.11

0.02

0.09

p=0.140

p=0.354

p=0.569

p=0.839

p=0.272

Mean speed of wind (m/s)

Pressure (hPa)

-0.19

-0.16

0.00

0.15

0.01

p=0.395

p=0.480

p=0.987

p=0.106

p=0.922

Total daily precipitation (l/m3)

Mean air temperature (°C)

-0.17

0.04

-0.16

-0.03

-0.039

p=0.461

p=0.855

p=0.392

p=0.746

p=0.631

Mean atmospheric humidity (%)

Mean speed of wind (m/s)

0.21

0.34

0.30

0.06

-0.08

p=0.353

p=0.116

p=0.106

p=0.528

p=0.346

Pressure (hPa)

-0.29

-0.37

-0.30

0.09

0.04

p=0.195

p=0.093

p=0.098

p=0.340

p=0.633

p - level of probability for significance of Pearson’s correlation coefficient

Lines 308-319:

3.7. Assessment of the number of admissions relative to the environmental variables, throughout the year in the consecutive months from January 2011 to December 2011.

The relationship of the number of cardiac events to the environmental variables co-occurring in the consecutive months throughout 2011 was examined in Table 10. No significant relationships between the number of cardiac events and the environmental factors were identified in any of the months (whether with or without halny wind) (p>0.05).

Table 10. Assessment of the number of cardiac events/ number of admissions relative to the environmental variables from January 2011 to December 2011.

Month

Environmental variables vs number of cardiac events

R2

Adjusted R2

F

p

January*

0.31

0.09

0.52

0.760

February

0.51

0.26

1.52

0.226

March*

0.46

0.21

1.36

0.274

April

0.41

0.17

1.04

0.415

May

0.54

0.29

2.04

0.107

June

0.21

0.05

0.23

0.945

July

0.39

0.15

0.93

0.476

August

0.50

0.25

1.64

0.187

September

0.44

0.19

1.15

0.360

October*

0.38

0.14

0.84

0.533

November*

0.29

0.08

0.44

0.818

December*

0.33

0.11

0.63

0.679

R2- regression model. Adjusted R2 – regression model disregards extreme values. F – result of Fisher’s test. p– significance level. *months with halny

Lines 321-408:

  1. Discussion

Based on a review of the literature it can be concluded that many researchers are investigating the association between wind-related environmental variables, and cardio- or cerebrovascular events, including the impact observed on days preceding onset of such weather conditions [14-23].

The current study was an attempt to collect evidence related to the impact of foehn wind and the related environmental variables on the incidence of cardiac events. The basic meteotropic factor adopted in the study was defined as halny wind and the accompanying change from cold to warm weather front.

Contrary to the findings reported by other researchers [14-23], the current study did not confirm effects of environmental factors investigated, i.e. air temperature, atmospheric humidity, speed of wind, pressure and total daily precipitation in the incidence of cardiac events, either on halny days, on days preceding the onset of halny as well as throughout the whole year in the consecutive months from January 2011 to December 2011. The findings showed no statistically significant correlations between the environmental variables and the number of cardiac events in any of the periods analysed (i.e. days with halny, days preceding halny onset, total days with and preceding halny, other days, all the days combined. This may be linked with the fact that the data acquired in the current study may be treated as preliminary, and in fact we are planning to carry out further study where we want to acquire more comprehensive data from other hospitals. However, it should be remembered that some of the hospitals may be located further away than others from the Tatra Mountains. Hence, it may be difficult to compare the acquired results. Likewise, Honda et al. explained that their study focused on the association between meteorological factors and frequency of acute myocardial infarction in patients residing in a small, strictly defined region, because the potential variations in weather conditions that are likely to occur in a larger area would have adversely affected their analyses [17]. In view of the above, in our study we have collected evidence on the effects of halny wind in this particular territory. Nevertheless, it is necessary to conduct further research which will take into account other factors, such as air pollution, individuals’ socioeconomic status, lifestyle, body mass index, physical activity, etc. because, based on our findings it may be hypothesised that since there are no correlations between the environmental variables and the number of cardiac events, incidence of myocardial infarction may be linked with other factors rather than weather. These issues will be investigated in our future studies. Other researchers also point out it is necessary to assess other risk factors that may influence the relationships of environmental variables and the incidence of cardiac events [14,20-22].

Although the findings covering the relevant period of time do not show significantly higher number of cardiac events on halny days, a greater number of the related hospital admissions was identified in the case of days immediately preceding occurrence of halny. On the days preceding halny wind onset, the incidence of cardiac events was higher than on the remaining days. However, the findings showed no statistically significant correlation between environmental variables and the number of cardiac events observed during this time. Halny wind frequently leads to a considerable increase in the temperature and decrease in humidity on the leeward side of the mountains. The strong abrupt gusts of wind at times reach a speed typical for hurricane [4]. Despite the fact that no significant correlations were observed between environmental variables and cardiac events on days immediately preceding halny onset, the graphs in Figures 1 and 2 clearly show that on the consecutive days from 1 to 14 December, which included seven days with halny, the mean temperature in most cases was lower on the day preceding halny, compared to the day with halny onset; atmospheric humidity in most cases was lower on the day preceding halny compared to the day with halny onset; similarly speed of wind on the days preceding halny in most cases was higher than or equal to the speed of wind on the day with halny onset. Hence, it appears that the weather conditions observed on the days preceding halny onset may exacerbate the risk of myocardial infarction, a conclusion which is consistent with opinions presented by other researchers. For example, Honda et al. found that increased incidence of acute myocardial infarction corresponds to lower minimum temperature occurring on the 2nd day preceding the onset of wind [17]. Similarly, Vencloviene et al. showed that prognostic factors for the risk of acute coronary syndromes include such environmental variables as wind speed, daily air temperature, and barometric pressure on the day of the onset and on the two days preceding the ambulance call [18]. Furthermore, Delyukov et al. and Głuszak et al. suggest that similar synchronicities can be observed during a period ranging from 3 to 24 hours prior to an arrival of a weather front [30,31]. Analysis of the mean pressure and the number of cardiac events in the consecutive days from 1 to 14 December, including 7 days with halny, shows that in most cases pressure was higher on the days preceding halny, compared to the days with halny onset, and the number of cardiac events was higher on the days preceding halny in six cases and only in one case was lower than on the day of halny onset. It can be speculated that the greater number of cardiac incidents identified in our study 24 hours before onset of halny wind may be associated with the fact that indeed the wind is only an effect of earlier turbulent changes in atmospheric pressure systems. The factor of key importance here may be the drop in atmospheric pressure since onset of halny is determined by co-occurrence of high pressure on the southern side and low pressure on the northern side of the Tatra mountains. Other published studies confirm that changes in atmospheric pressure produce adverse effects in human body, possibly also leading to cardiac incidents [33-36]. However, our study did not demonstrate statistically significant correlations between cardiac events and pressure, which may result from the short period of the observations. Therefore, in order to establish a cause-and-effect relationship and some temporal patterns, it is necessary to conduct a more comprehensive study taking into account more data from a larger number of hospitals and related to a longer period of time. By expanding the research to include additional years of data it may be possible to increase these numbers and further determine what role temporal effects play.

  1. Strengths and limitations of this study

This is the first exploratory article, the aim of which is to provoke discussion on the impact of foehn wind and the related environmental variables on the incidence of cardiac events in the population inhabiting a region affected by halny wind in Poland. The first study limitation lies in the fact that although the obtained data relate to 465 cardiology patients and cover a 365 day period, in fact the halny days and number of admissions on these days or days either side of halny days only account for a small portion of the total data. To establish a cause-and-effect relationship the relevant data should be acquired from a larger area affected by halny wind in Poland. The second limitation of the study is linked with the insufficient duration of the observation period, and the fact that the analyses took into account data related to one calendar year only; additional years of data would be helpful to increase these numbers and further determine what role temporal effects play. The third limitation results from the fact that we did not consider certain risk factors, such as air pollution and individuals’ socioeconomic status, lifestyle, body mass index and physical activity, that may influence the relationships of environmental variables on the incidence of cardiac events.

Lines 410-423:

Conclusions

  1. The study failed to demonstrate statistically significant correlations between the environmental factors and the number of cardiac indicants on the days with halny, as well as days preceding halny onset and the remaining days.
  2. On the days with halny wind, incidence of cardiac events is similar to that on the remaining days of the year.
  3. Overall, we believe that our preliminary results may contribute to the discussion concerning the impact of foehn wind and the associated environmental factors on the incidence of myocardial infarction in people living in the areas affected by halny wind in Poland. This analysis suggests that in order to establish a cause-and-effect relationship, more comprehensive data should be acquired from a larger number of regions affected by halny wind in Poland, and the relevant observations should be continued for a period longer than one calendar year. It will also be necessary to take into account effects produced by other factors, such as air pollution, individuals’ socioeconomic status, lifestyles, body mass index and physical activity.

Point 1: Minor remarks:

Line 96 shouldn’t have a question mark.

The x-axis legends in most of the plots seem incorrectly formatted (i.e. do not look like dates in the month of December).

Line 217: I don’t think “distributions” is the right word.

Response 2: Thank you for the helpful suggestions. In accordance with the Reviewer’s suggestion, we have removed the question mark, we have corrected the plots and have replaced “distributions” with “changes”.

BEFORE

Lines 95-96: We also decided to determine whether the environmental changes coincide with or predate the event examined?

Line 217: Distributions of mean daily air temperature, atmospheric humidity, speed of wind, pressure

AFTER

Lines 159: We also decided to determine whether the environmental changes coincide with or predate the event examined.

Line 288: Changes in mean daily air temperature, atmospheric humidity, speed of wind, pressure
